# Conditionally Site-Independent Neural Evolution of Antibody Sequences

Stephen Zhewen Lu [1 *]   Aakarsh Vermani [1 *]   Kohei Sanno [1]   Jiarui Lu [2 3]   Frederick A. Matsen IV [4 5 6]
Milind Jagota [7]   Yun S. Song [1]

## Abstract

Common deep learning approaches for antibody engineering focus on modeling the marginal distribution of sequences. By treating sequences as independent samples, however, these methods overlook affinity maturation as a rich and largely untapped source of information about the evolutionary process by which antibodies explore the underlying fitness landscape. In contrast, classical phylogenetic models explicitly represent evolutionary dynamics but lack the expressivity to capture complex epistatic interactions. We bridge this gap with CoSiNE, a continuous-time Markov chain parameterized by a deep neural network. Mathematically, we prove that CoSiNE provides a first-order approximation to the intractable sequential point mutation process, capturing epistatic effects with an error bound that is quadratic in branch length. Empirically, CoSiNE outperforms state-of-the-art language models in zero-shot variant effect prediction by explicitly disentangling selection from context-dependent somatic hypermutation. Finally, we introduce *Guided Gillespie*, a classifier-guided sampling scheme that steers CoSiNE at inference time, enabling efficient optimization of antibody binding affinity toward specific antigens.

## 1. Introduction

Antibodies are key effectors of the adaptive immune response, enabling humans and other vertebrates to recognize and neutralize an enormous diversity of molecular targets (antigens). This diversity is made possible by an accelerated evolutionary process operating within individuals, known as *affinity maturation*. During an immune response, B cells evolve in germinal centers, where their antibody genes undergo rapid somatic hypermutation (SHM) and are subsequently subjected to selection that favors variants with improved antigen binding. Repeated cycles of mutation and selection give rise to clonal trees of antibody variants, in which strongly binding lineages expand and diversify, while weak binders are progressively eliminated.

Affinity maturation thus represents a rare example of evolution that is both rapid and experimentally observable, making it an attractive system for studying how mutation and selection jointly shape protein function. To observe the results of this process, many studies have leveraged high-throughput sequencing to generate large-scale profiling of antibody repertoires from peripheral blood, yielding hundreds of thousands of antibody sequences per individual across diverse immune contexts. These data, rich in mature antibody sequences, have been used to train a plethora of antibody sequence models (Graves et al., 2020; Ruffolo et al., 2021; Leem et al., 2021; Olsen et al., 2022b; Hie et al., 2023; Kenlay et al., 2024b; Olsen et al., 2024a; Wang et al., 2025; Im et al., 2025; Kim et al., 2026). While such models have demonstrated impressive performance in capturing epistatic constraints within antibody sequences, they fundamentally lack the ability to model the time-dependent *process* over which antibodies mature. In particular, general antibody language models are trained under the assumption that the observed sequences in the clonal tree are *independent and identically distributed (i.i.d.)* samples from a stationary distribution. In reality, however, antibody sequences emerge through a structured evolutionary process as mutated descendants of specific germline progenitors. As a result, the performance of these models may in part stem from memorization of conserved germline residues rather than from a true understanding of the complex *process* of affinity maturation itself (Ng & Briney, 2025; Olsen et al., 2024a).

Developing principled and expressive models of sequence evolution that capture these dynamics remains a challenging task. From a theoretical standpoint, evolutionary processes are typically modeled using continuous-time Markov chains, but the immense state space of protein sequences renders direct modeling intractable. To mitigate this computational bottleneck, classical models of sequence evo-

---

[*] Equal contribution [1] University of California, Berkeley [2] Mila - Québec AI Institute [3] Université de Montréal [4] Fred Hutchinson Cancer Research Center [5] University of Washington [6] Howard Hughes Medical Institute [7] Columbia University. Correspondence to: Yun S. Song <yss@berkeley.edu>.

*Proceedings of the 43rd International Conference on Machine Learning*, Seoul, South Korea. PMLR 306, 2026. Copyright 2026 by the author(s).

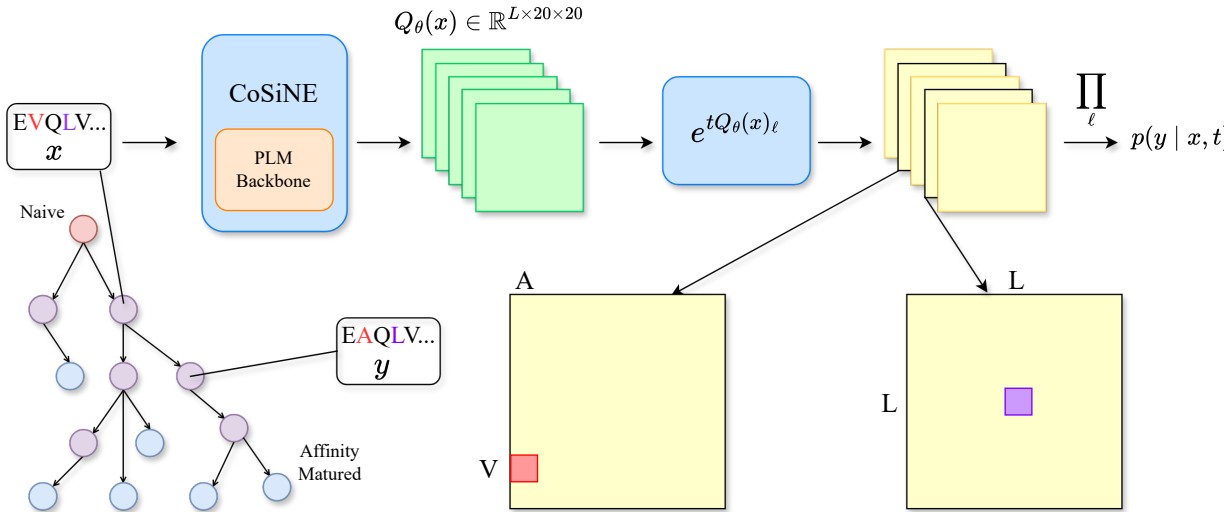

*Figure 1.* **Overview of CoSiNE**. Given an antibody sequence $x$, the neural network outputs site-specific rate matrices conditioned on the full sequence. Each matrix is evolved for duration $t$ to yield a per-site transition distribution $p(y_l \mid x, t)$. Assuming conditional independence, we take the product of the per-site transition probabilities to yield the full sequence transition probability $p(y \mid x, t)$.

lution assume that sites evolve independently according to a context-agnostic substitution process. While this assumption enables efficient likelihood computation and has underpinned decades of progress in phylogenetics, it comes at the cost of ignoring epistatic interactions between sites. As a consequence, these models have limited expressivity and often produce unrealistic evolutionary trajectories. This limitation has restricted their applicability in antibody sequence design and optimization, especially in comparison to modern antibody language models, which implicitly capture complex intra-sequence dependencies but lack explicit evolutionary grounding.

In this work, we aim to combine the strengths of these two paradigms by introducing a **co**nditionally **s**ite-**i**ndependent **n**eural **e**volution model (CoSiNE) that learns to simulate antibody affinity maturation while capturing epistatic interactions within the sequence (Figure 1). CoSiNE uses a neural network to parameterize site-specific rate matrices conditioned on the full sequence context, enabling a factorized transition likelihood that still captures dependencies among sites. In Section 2, we formalize the affinity maturation generative process and describe foundational CTMC theory for sequence evolution models. In Section 3, we review related work. In Section 4, we introduce the CoSiNE model and prove that it can learn epistatic effects by approximating an expressive sequential point mutation process. We fit CoSiNE to about a hundred thousand clonal trees and employ a principled Gillespie sampling algorithm to simulate clonal expansion. In Section 4.1, we describe how to disentangle the effects of selection and mutation on the affinity maturation process so that CoSiNE can be used to infer antibody fitness. In Section 4.2, we propose a classi-

fier guidance procedure to steer the functional properties of antibodies generated by CoSiNE. In Section 5, we demonstrate the effectiveness of our approach by outperforming existing models on zero-shot variant effect prediction and *in silico* antibody optimization tasks. Finally, in Section 6, we discuss limitations of the current framework and outline directions for future work.

## 2. Background

We consider the evolutionary history of antibody families, where each family $f$ is represented by a clonal tree $T_f$. At the root of $T_f$ is a naive antibody corresponding to an unmutated sequence derived from genetic recombination. Over time, this naive sequence accumulates somatic mutations, which are recorded in the tree as directed edges $(x, y)$ with an associated branch length $t \in \mathbb{R}^+$. Informally, we call $\tau = (x, y, t)$ an evolutionary transition from the *parent $x$* to the *child $y$* over *time $t$*. Following the standard in phylogenetics, we assume that $t$ is calibrated to the expected number of mutations per-site between $x$ and $y$.

### 2.1. Markov Models of Protein Evolution

Protein sequence evolution is most commonly modeled via continuous-time Markov chains (CTMCs). In general, given a discrete state space $\mathcal{S}$, a CTMC is completely defined by an initial distribution $p_0$ and a rate matrix $\mathbf{Q} \in \mathbb{R}^{|\mathcal{S}| \times |\mathcal{S}|}$. In fact, $\mathbf{Q}$ is all we need to compute the transition probability

$$p(y \mid x, t) := \exp(t\mathbf{Q})_{xy} \tag{1}$$

for maximum-likelihood estimation of $\mathbf{Q}$ as well as for most interesting sampling tasks. Unfortunately, tractability

is limited by the matrix exponential $\exp(t\mathbf{Q})$, which costs $O(|\mathcal{S}|^3)$ time. For protein sequences of length $L$ with the typical 20 amino acid vocabulary $\mathcal{A}$, this step is problematic for even $L > 2$ since $|\mathcal{S}| = |\mathcal{A}|^L = 20^L$.

## 2.2. Classical "Independent-Sites" Models

To circumvent this issue, classical models of protein evolution make the independent-sites assumption to enable the factorization of the full sequence Markov chain into $L$ independent chains, one for each site of the protein. Thus, instead of inferring a sequence-level transition matrix $\mathbf{Q} \in \mathbb{R}^{|\mathcal{A}|^L \times |\mathcal{A}|^L}$, classical models infer $Q_\ell \in \mathbb{R}^{|\mathcal{A}| \times |\mathcal{A}|}$ for $\ell \in \{1, \ldots, L\}$. The resulting transition probability is now tractable with a time complexity of $O(L|\mathcal{A}|^3)$:

$$p(y \mid x, t) := \prod_{\ell=1}^{L} \exp(tQ_\ell)_{x_\ell, y_\ell}$$

However, the independent-sites assumption leads to model misspecification because it fully ignores higher-order epistatic effects between sites. In fact, popular classical models like WAG (Whelan & Goldman, 2001) and LG (Le & Gascuel, 2008) learn a single $Q \in \mathbb{R}^{|\mathcal{A}| \times |\mathcal{A}|}$ shared by all $L$ sites, with an optional scaling factor $Q_\ell = \alpha_\ell Q$ to support site rate variation. As one can imagine, these models are under-expressive and fail to sample high quality proteins over long branch lengths. As such, their application in protein design and optimization has been limited.

## 2.3. Sequential Point Mutation Process

To capture epistatic effects, the class of Markov chains used to model protein evolution is commonly assumed to be a sequential point mutation process (Pedersen & Jensen, 2001; Gesell & von Haeseler, 2006). In this regime, we assume that mutations in the sequence are always introduced one at a time, prohibiting simultaneous events at multiple sites. This introduces a sparse structure to the rate matrix since its entries $\mathbf{Q}_{xy}$ can only be non-zero if $x$ and $y$ differ by a Hamming distance of 1. While this model ignores insertions and deletions (indels), it is a standard assumption in protein evolution that is particularly well-suited for antibodies. This is because somatic hypermutation is biologically driven by single-nucleotide substitutions (Noia & Neuberger, 2007), while indels are rare and typically filtered out by purifying selection to preserve antibody structure (Wu et al., 2003). We denote the space of such matrices by $\mathbf{Q} \in \mathbb{R}^{|\mathcal{A}|^L \times |\mathcal{A}|^L}$.

## 3. Related Work

To address the lack of site heterogeneity in classical models, Le et al. (2012) developed a mixture model that partitions sites into 4 rate categories and estimates a different substitution matrix for each partition. In the extreme case, Prillo

et al. (2024) estimated a separate rate matrix for each site. Their model, SiteRM, achieved strong variant effect prediction results, outperforming many protein language models on the ProteinGym benchmark (Notin et al., 2023).

To move beyond the independent-sites assumption, a common approach consists of conditioning the parameterization of site specific mutation rates on the surrounding sequence context. This idea was first explored by Felsenstein & Churchill (1996) who used a Hidden Markov Model to assign per-site rate categories depending on adjacent sites. Bérard et al. (2008) proposed a neighbor-dependent DNA substitution model that adds extra terms to a baseline rate matrix in the presence of CpG sites. More recently, Albors et al. (2025) used a neural network that inputs an entire DNA sequence and outputs the parameters of an F81 model (Felsenstein, 2005) for the central position of this sequence. Alternatively, Koehl et al. (2026) abandons the Markov chain formulation entirely and models evolutionary transitions as a machine translation task.

In the domain of antibodies, a separate line of work seeks to mitigate germline bias in protein language models through specialized training objectives or architectural modifications. Olsen et al. (2024b) proposed a focal loss that emphasizes mutated sites. Im et al. (2025) introduced a mutation-aware vocabulary that distinguishes between germline and mutated residues. Concurrent with our work, Kim et al. (2026) developed PRISM, which trains an auxiliary head to predict germline vs. non-germline status at each site, conditioning the language modeling head on this prediction. While these methods reduce germline bias, they collapse evolution to a binary germline-vs-mutated distinction injected through auxiliary losses, vocabularies, or prediction heads. A CTMC formulation models the underlying continuous-time process directly, with the germline-vs-mutated distinction emerging as a natural byproduct.

Most relevant to our work is the Deep Amino acid Selection Model (DASM) (Matsen et al., 2026), a mutation-selection model that disentangles somatic hypermutation (SHM) bias from functional selection. Trained on evolutionary transitions, DASM factorizes each codon transition likelihood into a fixed neutral SHM model and a learned amino-acid selection factor. However, the DASM transition likelihood requires a manual clamping of selection scores to remain a valid probability distribution. In contrast, COSINE derives a selection score through a log-likelihood ratio between a neural CTMC and a pre-trained SHM model (Sung et al., 2025). This formulation ensures mathematical consistency without heuristic constraints, enabling COSINE to model the affinity maturation process with greater accuracy. We provide a detailed discussion of our model against DASM in Section A.3.

# 4. COSINE: Conditionally Site-Independent Neural Evolution Model

To address the limitations of classical models, we developed COSINE, which introduces two key modifications that mitigate model misspecification and leverage high-dimensional neural parameterization. First, we decouple rate estimation by learning site-specific rate matrices $Q_\ell$ for each site $\ell$ instead of a single unified matrix with scaling factors. Second, we model each evolutionary transition $\tau = (x, y, t)$ with its own set of rate matrices $\{Q_\ell^{(\tau)}\}_{\ell=1}^L$, which are inferred by conditioning on the parent sequence $x$. Concretely, the transition probability of COSINE is given by

$$p_\theta(y \mid x, t) = \prod_{\ell=1}^L \exp\left(t Q_\theta(x)_\ell\right)_{x_\ell, y_\ell} \qquad (2)$$

where $Q_\theta : \mathbb{R}^{|\mathcal{A}|^L} \to \mathbb{R}^{L \times |\mathcal{A}| \times |\mathcal{A}|}$ is a function parameterized by a neural network. Intuitively, COSINE relaxes the standard independent-sites assumption by estimating state-conditional rates, which enables the model to learn epistatic effects. In the next section, we provide theoretical grounding for COSINE by showing that our model constitutes a first-order approximation of a sequential point mutation process over the full sequence space. We further justify this framework by establishing an upper bound for the error between the transition probability of our model and the transition probability of the point mutation process $P(y \mid x, t) = \exp(t\mathbf{Q})_{xy}$. Finally, we propose to sample from COSINE using a Gillespie procedure that provably samples from $P(y \mid x, t)$ under certain conditions.

**COSINE is a First-Order Approximation of Q.** Let $\mathbf{Q} \in \mathbb{R}^{|\mathcal{A}|^L \times |\mathcal{A}|^L}$ be the generator rate matrix of a sequential point mutation process over protein sequences of length $L$ with amino acid vocabulary $\mathcal{A}$. The vector of transition probabilities to other states after some time $t$ is given by indexing the transition matrix as follows:

$$P(\cdot \mid x, t) = \left(e^{t\mathbf{Q}}\right)_{x,\cdot} \in \mathbb{R}^{|\mathcal{A}|^L}$$

**Proposition 4.1.** *(Proof in Section D.1) Assume the per-site rate matrices $Q_\theta(x)_\ell$ are parameterized such that*

$$(Q_\theta(x)_\ell)_{x_\ell, y_\ell} = \mathbf{Q}_{x,y}$$

*for all $x, y$ with Hamming distance $d(x, y) = 1$ and $\ell$ is the unique site where $x$ and $y$ differ. Then, the error between the transition probability vectors is bounded such that*

$$\|P(\cdot \mid x, t) - p_\theta(\cdot \mid x, t)\|_1 \leq (\lambda t)^2 = O(t^2) \qquad (3)$$

*where $\lambda = \max_x\{-\mathbf{Q}_{x,x}\}$ is the maximum exit rate of any given state.*

Intuitively, the theorem shows that our model offers a principled way of capturing first-order mutation dynamics while restricting the approximation error to epistatic effects that grow quadratically in the branch length. Moreover, this approximation is particularly well-suited for affinity maturation because high-frequency clonal selection typically constrains lineages to short branch lengths. In this regime, the first-order evolutionary signal captured by our model $O(\lambda t)$ dominates the quadratic approximation error $O(\lambda^2 t^2)$.

**Gillespie Sampling with COSINE.** Although the factorized transition probability in Equation (2) allows for trivial sampling, it incurs an error with respect to the more expressive sequential point mutation process that scales quadratically with the branch length. To address this limitation, we propose a Gillespie sampling algorithm adapted to the instantaneous rates of the COSINE model. In Lemma 4.2, we show that this procedure provably samples from $P(\cdot \mid x, t)$ under certain conditions.

**Lemma 4.2.** *(Proof in Section D.2) Let $x_0, \ldots, x_{t_{N-1}}, x_{t_N}$ be the trajectory of sequences sampled from the Gillespie Algorithm S1, using branch length $t$ and starting sequence $x_0$. For all $x \in \{x_0, \ldots, x_{N-1}\}$, assuming that*

$$(Q_\theta(x)_\ell)_{x_\ell, y_\ell} = \mathbf{Q}_{x,y}$$

*holds for all sequences $y$ with Hamming distance $d(x, y) = 1$, then $x_{t_N} \sim P(\cdot \mid x_0, t)$.*

The validity of Proposition 4.1 and Lemma 4.2 relies on $(Q_\theta(x)_\ell)_{x_\ell, y_\ell}$ approximating the instantaneous non-zero rates in the point mutation process rate matrix $\mathbf{Q}_{x,y}$. However, because COSINE is trained with the approximate transition probability in Equation (2), its estimated rates will incur some bias due to the marginalization of unobserved intermediate states on long branches. We acknowledge this theoretical limitation and study its effect thoroughly in Section C.1. In practice, we show that the transition probability error of our model indeed scales quadratically with the branch length and that Gillespie sampling produces final state distributions that match those of the true process much more closely (Sections C.2 and C.3).

## 4.1. Inferring Antibody Fitness Landscapes from Affinity Maturation

Beyond accurately simulating affinity maturation, it is of great interest to explicitly learn the fitness of a given antibody sequence to enable the design of antibodies with desirable properties. The fitness of an antibody sequence is the objective that guides the affinity maturation process, which is typically some combination of properties including expressibility, binding affinity for a target epitope, and lack of affinity for self-epitopes. These properties are also desirable for artificially engineered antibodies.

A common method for determining an antibody model's ability to predict fitness is through zero-shot Variant Effect Prediction (VEP) with Deep Mutational Scanning (DMS) assays. In these assays, a library of mutants is produced from a wildtype antibody sequence, and each mutant is evaluated for a certain property, such as binding affinity for a specific epitope or level of expression.

For antibody language models that learn the marginal distribution of sequences $p(x)$, the most common approach to evaluating on DMS datasets is to correlate the sequence perplexity from the model with the true fitness of the sequence, under the assumption that lower perplexity (corresponding to higher predicted likelihood) corresponds with higher fitness. COSINE, however, learns a conditional distribution $p(y|x, t)$, requiring a different approach to DMS evaluation.

**Somatic Hypermutation Models.** Molecular evolution, including somatic evolution processes such as affinity maturation, can be viewed as a two-step process: First, mutations are introduced by an underlying mutational mechanism. Second, these mutations are filtered by selection according to the fitness advantages or disadvantages they confer.

It is generally believed that mutations arise independently of natural selection. During antibody affinity maturation, this underlying process is known as somatic hypermutation (SHM). Machine learning models, such as Thrifty (Sung et al., 2025), have been developed to predict rates of SHM from surrounding context by training on sequences that have undergone frameshift mutations, meaning they are under little or no selection, preventing any confounding.

**Disentangling Selection from SHM.** To estimate the fitness landscape, we treat the underlying SHM process as the baseline. Following the mutation-selection framework of Halpern & Bruno (1998), we decompose $Q_{xy}$, the observed transition rate from sequence $x$ to sequence $y$, as:

$$Q_{xy} = k \, \mu_{xy} \, P_{\text{fix}}(x \to y), \qquad (4)$$

where $\mu_{xy}$ is the transition rate under SHM, $P_{\text{fix}}(x \to y)$ is the probability of fixation, and $k$ is an arbitrary scalar.

Using the small $t$ approximation $p_\theta(y \,|\, x, t) = \exp(tQ_\theta)_{xy} \approx t(Q_\theta)_{xy}$ for $x \neq y$, we can manipulate Equation (4) to derive the following *selection score* that we use to evaluate COSINE on DMS assays:

$$\text{Score}(x \to y) = \log p_\theta(y \mid x, t) - \log q(y \mid x, t) \quad (5)$$
$$\approx \log\left(P_{\text{fix}}(x \to y)\right) + C,$$

where $q(y \mid x, t)$ is the probability of sequence $x$ mutating to $y$ under Thrifty (more details on how this is calculated in Section B.3.2) and $p_\theta(y \mid x, t)$ is the probability of the transition under COSINE. In Section D.3, we rely on standard population genetics theory (Kimura, 1962) to show that

$P_{\text{fix}}(x \to y)$ is monotonic with respect to $s_{xy}$, the selective advantage of allele $y$ over allele $x$.

## 4.2. Conditional Sequence Optimization via Guided Gillespie Sampling

Biologists often wish to design antibodies that strongly bind to a target of interest while maintaining expressibility and stability in the human body. While we are able to show that our model learns a very strong prior for the latter properties, there remains a performance gap for target-specific properties such as binding affinity. Indeed, given that the antigens associated with the transitions in our dataset are unknown, it is possible that the selection scores inferred by our model (Equation (5)) do not align with wet lab measurements for binding affinity to an arbitrary target. Instead of fine-tuning COSINE with additional expensive experimental data, we propose a classifier guidance approach to sample from the desired posterior transition density $p(y \mid x, t, z)$ conditioned on the antigen of interest $z$.

Nisonoff et al. (2025) showed that as we take the limit of $t \to 0$, the above posterior transition density is completely defined by the following generator rate matrix:

$$(\mathbf{Q}_z^{(\gamma)})_{x,y} = \left[\frac{p(z \mid y)}{p(z \mid x)}\right]^\gamma \mathbf{Q}_{x,y}, \qquad (6)$$

where $\gamma \in \mathbb{R}^+$ is a hyperparameter that controls the guidance strength and $p(z \mid y)$ is the marginal likelihood of the antigen $z$ given the selection of antibody $y$. Learning $p(z \mid y)$ for an arbitrary class of high-dimensional antigen sequences is difficult; therefore, we approximate it using a surrogate likelihood for the measured binding affinity between $y$ and $z$. Specifically, we model the probability of the antigen being $z$ as proportional to the probability that the affinity $r$ exceeds a threshold $r_0$. Furthermore, we assume that the affinity $r$ between $y$ and $z$ is normally distributed with mean $\mu_{\theta_z}(y)$ and variance $\sigma_{\theta_z}^2(y)$, such that

$$p(z \mid y) \propto \mathbb{P}(r > r_0 \mid y; \theta_z) = \Phi\left(\frac{\mu_{\theta_z}(y) - r_0}{\sigma_{\theta_z}(y)}\right), \quad (7)$$

where $\Phi$ denotes the c.d.f. of the standard normal distribution. Instead of fixing a global value for $r_0$, which could lead to vanishing guidance weights in regions where sequences are far from the threshold, we select $r_0$ adaptively by setting it to the parent's mean prediction $r_0 = \mu_{\theta_z}(x)$. Under the assumption of a symmetric predictive distribution (like the Gaussian used here), the denominator $\mathbb{P}(r > \mu_{\theta_z}(x) \mid x, \theta_z)$ is exactly $1/2$. Consequently, Equation (6) becomes

$$(\mathbf{Q}_z^{(\gamma)})_{x,y} = \left[2 \cdot \Phi\left(\frac{\mu_{\theta_z}(y) - \mu_{\theta_z}(x)}{\sigma_{\theta_z}(y)}\right)\right]^\gamma \mathbf{Q}_{x,y}. \quad (8)$$

In its current form, Equation (8) is very expensive to compute since we need to query $\mu_{\theta_z}(y)$ and $\sigma_{\theta_z}(y)$ for all se-

quences $y$ with Hamming distance $d(x, y) = 1$, which corresponds to $L \times (|\mathcal{A}| - 1) + 1$ calls to the predictor for each $x$. Instead, we employ a first-order Taylor series approximation about $\mu_{\theta_z}(x)$ while assuming a locally constant variance: $\sigma_{\theta_z}(y) \approx \sigma_{\theta_z}(x)$. By linearizing the mean as $\mu_{\theta_z}(y) \approx \mu_{\theta_z}(x) + \nabla_x \mu_{\theta_z}(x)^\top (y - x)$, we can estimate the change in fitness for all $y$ via a single gradient computation. Substituting these approximations into Equation (8) yields the Taylor-approximated guidance (TAG) form:

$$(\mathbf{Q}_z^{(\gamma)})_{x,y} \approx \left[ 2 \cdot \Phi \left( \frac{\nabla_x \mu_{\theta_z}(x)^\top (y - x)}{\sigma_{\theta_z}(x)} \right) \right]^\gamma \mathbf{Q}_{x,y}. \quad (9)$$

We apply TAG to Algorithm S1 to obtain *Guided Gillespie* sampling and provide the pseudocode in Algorithm S2. Although *Guided Gillespie* follows from Nisonoff et al., our formulation is applied to the rate matrix $\mathbf{Q}$ of any sequential point mutation process, which is not constrained by boundary time conditions like the generators of flow matching or discrete diffusion models. Notably, this implies that our predictor does not need to be trained on noisy sequences, enabling the straightforward application of sequence-to-property predictors trained naively on experimental data.

## 5. Experiments

### 5.1. Fitting CoSINE on a Clonal Tree Dataset

We fit CoSINE on a dataset of $\sim$2 million evolutionary transitions constructed from 5 public sources (Jaffe et al., 2022; Tang et al., 2022; Vergani et al., 2017; Engelbrecht et al., 2025; Rodriguez et al., 2023), with train and test splits that match those of the DASM model (Matsen et al., 2026). We initialized CoSINE from the 150M parameter ESM2 checkpoint (Lin et al., 2023) and replaced the language modeling head with a randomly initialized output head that uses the `softplus` activation function to estimate the off-diagonal rates of $Q_\theta(x)_\ell$. To satisfy the properties of a valid rate matrix, we set the diagonal entries of $Q_\theta(x)_\ell$ to the negative sum of their respective rows. We also inserted a chain-break token between the heavy and light sequences of paired antibodies, enabling simultaneous reasoning over both chains. We performed end-to-end training of all parameters using a polynomial decay learning rate schedule and early stopping based on validation loss (Additional details in Appendix A).

We found that CoSINE fits this data remarkably well, achieving a test perplexity of 1.264 for heavy chain transitions in the Rodriguez et al. dataset (see Appendix A for our perplexity definition). In a head-to-head comparison with DASM+Thrifty, CoSINE fits the test transitions with greater per-site likelihood in 62.3% of cases (Figure 2). Interestingly, this improvement is even more significant for long branch lengths ($t > 0.25$).

To isolate the contribution of context-conditioning, we also

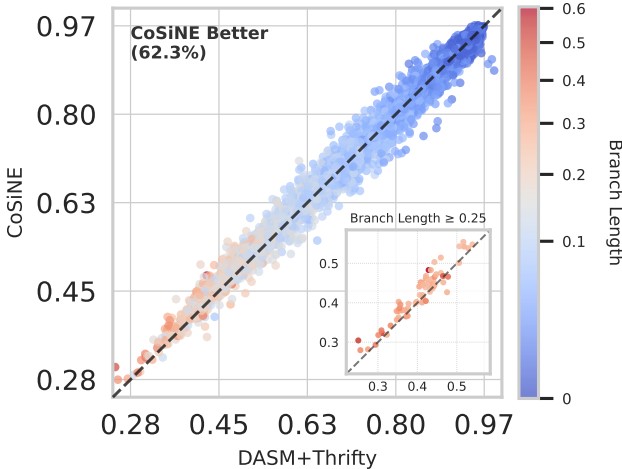

*Figure 2.* **Mean per-site likelihood of CoSINE versus DASM+Thrifty on held out evolutionary transitions from the test set.** CoSINE achieves better model fit, especially on transitions with longer branch lengths ($t \geq 0.25$).

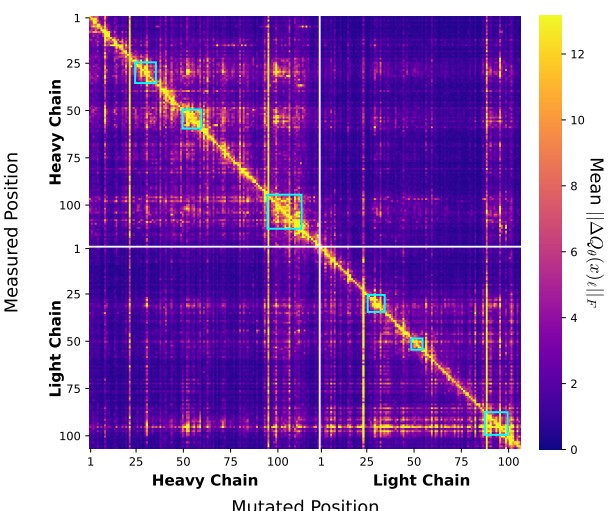

*Figure 3.* **Categorical Jacobian for antibody 47D11 from CoV-AbDab.** The heatmap displays the sensitivity of the model's output predictions (y-axis) to specific mutations in the input sequence ($x$-axis). Sensitivity is measured as the Frobenius norm of the change in predicted rate matrix, averaged over all possible mutations. **Cyan squares** denote the CDR regions for both chains.

compare against a classical context-independent baseline: a WAG-style exchangeability matrix fit on the same antibody training data. CoSINE achieves lower per-token NLL across all branch lengths, with the gap widening for longer branches (Section C.5).

### 5.2. CoSINE Captures Intra- and Inter-Chain Epistasis

To investigate the epistatic effects learned by CoSINE, we calculate the categorical Jacobian (Figure 3), inspired by

Zhang et al. (2024), for an arbitrary antibody sequence from the CoV-AbDab database (Raybould et al., 2020). Specifically, for each possible single amino acid substitution in the input sequence, we measure the change in the output at all positions (See Algorithm S3).

The majority of the measured effect is found along the diagonal. This is expected since changes in a given site and its immediate neighbors will likely have a major impact on its rate matrix due to the importance of local interactions. However, there is a considerable amount of off-diagonal activity as well. For example, we see that within the CDR regions, denoted by cyan squares, there is a high degree of off-diagonal coupling. This is biologically plausible, as the CDR regions combine to form the antigen-binding pocket of the antibody, so changes in one CDR residue are often correlated with changes in other CDR residues to preserve binding geometries. In fact, we see significant changes in the predictions at CDR1 and CDR2 caused by mutations in CDR3 within both the heavy and light chains. CoSiNE also detects this dependence across chains. For example, when introducing mutations to light chain CDR3, three distinct sensitivity hotspots are induced in the heavy chain associated with the three CDR regions.

### 5.3. Zero-Shot Variant Effect Prediction

We evaluate CoSiNE's performance on DMS assays by measuring the Spearman correlation of our selection score (Equation (5)) with the experimental fitness values. To calculate the score, we set the parent $x$ to be the wildtype sequence from which the mutants in the assay are derived. Rather than tuning $t$ per assay, we fix $t = 0.2$ for all assays and VEP experiments.

Evaluation is performed on four DMS assays, two measuring expression and two measuring binding, taken from the FLAb2 benchmark (Chungyoun & Gray, 2025). These assays were selected for their large sample sizes of sequences with the same length in an attempt to limit spurious correlations. As baselines, we compare against AbLang-2, DASM, PRISM, ProGen2 Small, ProGen2 Medium (best performing model for all FLAb2 binding datasets), ESM-2 150M (best performing model for all FLAb2 expression datasets), and ESM-2 650M (Olsen et al., 2024b; Matsen et al., 2026; Kim et al., 2026; Nijkamp et al., 2022; Lin et al., 2023). Further details on the DMS assays and evaluation with baseline models are provided in Section B.3.1.

The VEP results are shown in Table 1, where CoSiNE outperforms all other baselines on all datasets except Koenig Expression Light Chain, where it falls short of ProGen2 Small by 0.005. Interestingly, CoSiNE substantially outperforms all other models on the Adams dataset, where the wildtype sequence is a mouse antibody. On two additional binding assays from Petersen et al. (2024), using distinct an-

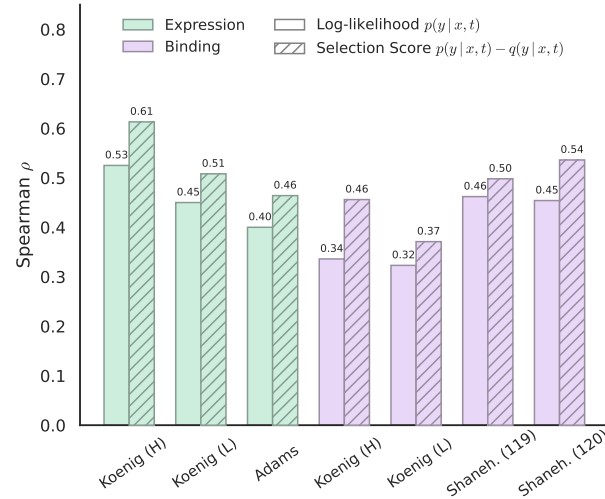

*Figure 4.* **DMS evaluation results for CoSiNE across expression (green) and binding (purple) assays.** A solid color indicates the log-likelihood, $\log p_\theta(y \mid x, t)$, and hatching indicates the selection score defined in Equation (5), which utilizes Thrifty likelihoods to separate selection from neutral mutation.

tibodies, antigens, and the MAGMA-seq assay technology, CoSiNE again outperforms all baselines (Table S4).

We investigate the efficacy of our approach to correcting for SHM on zero-shot VEP performance. As shown in Figure 4, our selection score leads to increased correlation with fitness across all datasets. Figure S6 provides an intuitive picture of how the SHM correction helps isolate a selection signal from CoSiNE. We also see that the ability to model interchain interactions is important. In Section C.8, we ablate this ability by passing in just the heavy or light chain alone, finding that the selection score's correlation with fitness drops substantially for some datasets.

Two additional experiments support the robustness of these results. First, CoSiNE's selection score is stable across a wide range of branch lengths, with $\Delta\rho \leq 0.045$ for $t \in [0.1, 0.4]$ (Section C.9). Second, training CoSiNE from scratch without the ESM2 backbone reduces performance by only $\Delta\rho = 0.041$ on average, indicating that the evolutionary training objective, rather than the ESM2 initialization, drives much of the predictive power (Section C.10).

### 5.4. Guided Affinity Maturation from Naive Antibodies

We demonstrate the potential of CoSiNE to simulate affinity maturation towards high-affinity binders, starting from naive antibody sequences. We used predictors from (Jin et al., 2022) trained on the CoV-AbDab database, which employ an RNN encoder to transform heavy chain sequences into neutralization scores $\mu_{\theta_z}$ against the SARS-CoV-1 and SARS-CoV-2 receptor binding domains. We adopted MC dropout (Gal & Ghahramani, 2015) to estimate $\sigma_{\theta_z}$. From

*Table 1.* **Comparison of deep protein models on VEP benchmarks across expression and binding landscapes as measured by Spearman correlation.** The Koenig assays are split depending on which chain is mutated, denoted by (H) and (L) for the heavy and light chains respectively. To maintain the same sequence length across samples, the Shanehsazzadeh (Shaneh.) assay is split into two datasets with heavy chains of lengths 119 and 120. Best performing models are shown in **bold**; second-best are underlined.

| MODEL | EXPRESSION | | | BINDING | | | |
|---|---|---|---|---|---|---|---|
| | KOENIG (H) | KOENIG (L) | ADAMS | KOENIG (H) | KOENIG (L) | SHANEH. (119) | SHANEH. (120) |
| ABLANG-2 | 0.096 | −0.127 | −0.097 | −0.090 | −0.011 | 0.253 | 0.209 |
| DASM | 0.596 | 0.474 | 0.270 | 0.415 | 0.327 | 0.450 | **0.536** |
| PRISM | 0.069 | 0.129 | 0.297 | 0.005 | 0.061 | 0.348 | 0.286 |
| ESM2-150M | 0.413 | 0.485 | −0.112 | 0.112 | 0.266 | 0.236 | 0.205 |
| ESM2-650M | 0.326 | 0.429 | 0.124 | 0.063 | 0.265 | 0.227 | 0.360 |
| PROGEN2-SMALL | 0.407 | **0.513** | −0.024 | 0.098 | 0.332 | 0.119 | 0.070 |
| PROGEN2-MEDIUM | 0.392 | 0.408 | 0.231 | 0.085 | 0.235 | 0.299 | 0.319 |
| COSINE (OURS) | **0.613** | 0.508 | **0.464** | **0.456** | **0.371** | **0.498** | **0.536** |

here, we randomly picked naive sequences from the OAS database (Olsen et al., 2022a) and recursively sampled down the tree in Figure S1 using *Guided Gillespie*.

We compare the affinity gain of our generated leaf sequences against 415 SARS-CoV-1 and 766 SARS-CoV-2 binders from the Cov-AbDab database. As shown in Figure 5, unguided sampling yields a distribution centered near zero for the SARS-CoV-1 target, confirming that the base model is unbiased regarding antigen specificity. Introducing guidance consistently shifts this distribution towards higher affinity. While $\gamma \geq 10$ produced scores exceeding biological binders (likely exploiting oracle uncertainty), we noticed that $\gamma = 5$ generates affinity profiles that overlap with the real binders. We therefore adopted $\gamma = 5$ for further evaluation. Figure 6 demonstrates that these guided samples maintain structural plausibility (AbodyBuilder3 pLDDT, Kenlay et al. (2024a)) and humanness (OASis, Prihoda et al. (2021)) comparable to both unguided and natural sequences. These results hold across both targets and many seed sequences (Section C.13), underscoring the generalizability of our approach.

Computationally, our first-order Taylor approximation achieves a 500-fold speedup over exact oracle guidance, with no significant difference in fitness improvements (Section C.11). The resulting runtime scales linearly in both sequence length and number of sampling steps (Section C.12).

### 5.5. Local Optimization of Antibody CDRs

Beyond long-range affinity maturation, we evaluate whether COSINE can perform constrained local optimization of antibody CDRs. We tasked each method with refining a SARS-CoV-1 binder under a strict budget of 5 mutations restricted to CDR positions, using the same neutralization predictor as in Section 5.4. For COSINE, we fix the number of Guided Gillespie steps to respect this ceiling. All budget-constrained methods are further limited to at most 5 oracle calls per sampled variant; we benchmark COSINE against a genetic algorithm (GA), a Product-of-Experts (PoE) sam-

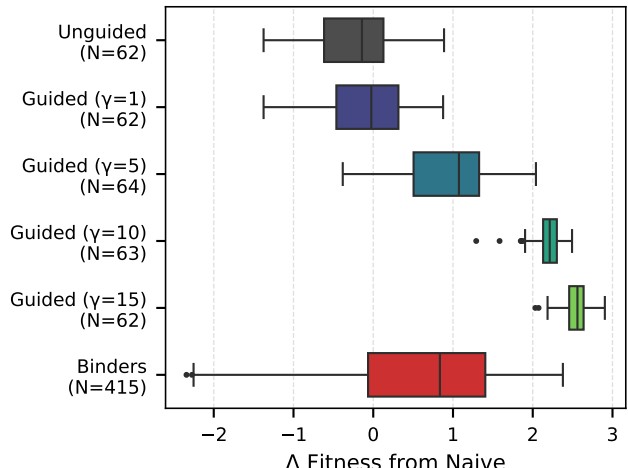

*Figure 5.* ***Guided Gillespie* consistently steers the predicted binding affinity of sampled leaf sequences.** We plot the change in predicted binding affinity from the naive root sequence used to start sampling. Known SARS-CoV-1 binders are in red.

pler (Gordon et al., 2024) applied to ESM-2 and AbLang-1, and an unbounded greedy hill-climbing baseline as a reference upper bound.

Generating 1,000 variants per method, we evaluate across binding affinity ($\Delta$Bind), sample diversity, and humanness (OASis). COSINE attains the highest mean and maximum $\Delta$Bind among the budget-constrained methods while preserving humanness comparable to the antibody-specialized AbLang-PoE baseline (Table 2). The PoE methods obtain slightly higher diversity but lower binding gains. Full baseline configurations are provided in Section B.5.

## 6. Conclusion and Limitations

In this work, we presented COSINE, a method that reconciles the expressivity of deep protein language models with the temporal dynamics of phylogenetic substitution models

*Table 2.* **Constrained CDR optimization of a SARS-CoV-1 binder.** Each variant is limited to 5 mutations within the CDR regions and is generated using at most 5 oracle calls. GREEDY* is exempt from the oracle-call budget and serves as a reference upper bound. $E_{\text{dist}}$ reports the mean edit distance from the seed sequence; INTDIV reports the mean pairwise edit distance between generated variants; $N_{\text{oracle}}$ reports the average number of oracle calls per generated sequence. Best results are **bolded**; second-best underlined.

| METHOD | UNIQUE ↑ | $E_{\text{DIST}}$ | INTDIV ↑ | OASIS ↑ | $\Delta$ BIND$_{\text{MEAN}}$ ↑ | $\Delta$ BIND$_{\text{MAX}}$ ↑ | $N_{\text{ORACLE}}$ ↓ |
|---|---|---|---|---|---|---|---|
| GREEDY* | 0.996 | 5.00 | 6.21 | 0.807 | 0.385 | 0.466 | 2756 |
| GENETIC ALGO. | 0.991 | 4.47 | 7.30 | 0.789 | 0.196 | 0.363 | 4.81 |
| ABLANG-1-*PoE* | 0.999 | 4.97 | 7.71 | **0.849** | 0.167 | 0.390 | 0.58 |
| ESM2-*PoE* | **1.000** | 4.98 | **8.47** | 0.802 | 0.149 | 0.355 | 0.58 |
| COSINE (OURS) | 0.988 | 4.56 | 6.83 | 0.822 | **0.198** | **0.395** | 5.00 |

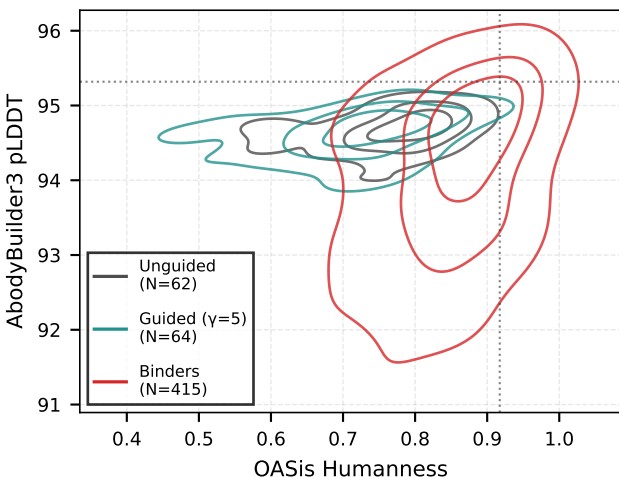

*Figure 6.* **Antibodies sampled by COSINE under guidance ($\gamma = 5$) maintain high structural quality (AbodyBuilder3 pLDDT) and humanness (OASis).** We compare against unguided leaf samples (black) and CoV-AbDab binders (red).

to effectively model the process of antibody affinity maturation. We introduced a mathematically grounded framework for the inference of sequential point mutation CTMCs via a first-order approximation of the true transition likelihood with bounded error. Empirically, we demonstrated the utility of COSINE by achieving state-of-the-art results in antibody variant effect prediction. Furthermore, to our knowledge, we are the first to draw an explicit connection between discrete diffusion and classical sequence evolution models, enabling the application of classifier guidance to steer the sampling process of our model. Our results demonstrate that it is possible to capture both time-dependent evolution and epistatic interactions, leading to a new paradigm for protein sequence design that is grounded in molecular evolution.

Despite these contributions, our approach has limitations that present directions for future work. First, our reliance on a first-order approximation of the CTMC over the full sequence state space leads to model misspecification. Second, our framework currently ignores insertions and deletions, restricting COSINE to equal-length sequences or multiple sequence alignments. Although these limitations are less

worrisome for modeling antibody affinity maturation, where evolution is rapid and indels are scarce, they likely present a greater challenge when generalizing to all types of proteins. Nonetheless, we believe COSINE provides a strong foundation for future research into dynamic and expressive generative models of protein evolution.

## Reproducibility Statement

We provide implementation and experimental details in Appendices A and B. The theoretical results in the main text are derived or proven in Appendix D. The source code for COSINE is made publicly available at `https://github.com/thematrixmaster/cosine`, including any datasets we used in this study.

## Acknowledgements

We would like to acknowledge Antoine Koehl and Kevin Sung for their insightful feedback and help with data preprocessing. This work was supported in part by the UC National Laboratory Fees Research Program of the University of California Office of the President (UC AI Science at Scale Grant L26CR10102), and NIH grants R01-HG013117 (Song) and R01-AI146028 (Matsen). Frederick Matsen is an investigator of the Howard Hughes Medical Institute.

## Impact Statement

This work advances machine learning by developing principled models for learning and simulating biological sequence evolution. By bridging deep learning with classical evolutionary modeling, our approach contributes new tools for understanding antibody affinity maturation and, more broadly, molecular evolution. Potential positive impacts include improved methods for studying immune responses and supporting applications such as vaccine design and therapeutic antibody development. These advances could contribute to better diagnostics and treatments. We view this work as a methodological contribution to machine learning and computational biology, with societal implications comparable to prior advances in biological sequence modeling.

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

# A. Additional Details of CoSINE

## A.1. Data Collection and Model Training

To train the CoSINE model, we compiled B-cell receptor (BCR) sequencing datasets from five sources (Jaffe et al., 2022; Tang et al., 2022; Vergani et al., 2017; Engelbrecht et al., 2025; Rodriguez et al., 2023). We adopted the data processing and phylogenetic inference protocol described by (Matsen et al., 2026). Briefly, sequences were clustered into clonal families and naive germlines were inferred using `partis` (Ralph & Matsen IV, 2022). We retained families with at least two productive sequences, defining productivity by the absence of stop codons and the presence of canonical cysteine and tryptophan codons flanking the CDR3 in the same reading frame as the V segment start. Consistent with recent large language model training pipelines (e.g., AbLang-2), we further excluded sequences with mutated conserved signature cysteines. Insertions and deletions were reversed to align all sequences to their naive ancestor without gaps.

For phylogenetic reconstruction, we performed tree inference and ancestral sequence reconstruction (ASR) using IQ-TREE (Nguyen et al., 2014) under the K80 substitution model, using the naive sequence as an outgroup. We accounted for mutation rate heterogeneity across sites via a 4-category FreeRate model. For paired heavy and light chain data, we specifically employed the edge-linked-proportional partition model to allow the chains to evolve at distinct overall rates. This pipeline yielded a final training set of parent-child pairs (PCPs) extracted from the edges of the resulting phylogenetic trees, comprising approximately $\sim$2 million transitions from $\sim$120,000 clonal families collected from 555 individual donors. We defer further details to the Matsen et al. paper.

Models were trained with mixed precision (BF16) for a maximum of 1 million steps, using a batch size of 16 with gradient accumulation over 3 steps. We employed the AdamW optimizer with a learning rate of $2.5 \times 10^{-4}$, utilizing 5,000 warmup steps followed by a polynomial decay schedule with a power of 2.0. To prevent overfitting, we applied a weight decay of 0.01 specifically to parameters with two or more dimensions (e.g., weights and embeddings), while excluding biases and layer normalization parameters. Gradients were clipped at a norm of 1.0, and training employed early stopping with a patience of 50 intervals based on validation loss. In practice, the loss converged after about 1 day on a single A100 GPU.

## A.2. Gillespie and *Guided Gillespie* Sampling

We provide the pseudocode for both unconditional Gillespie sampling and guided Gillespie sampling using CoSINE. Differences for the guided version are highlighted in red. Notice that only a single evaluation of the predictor is required per step.

---

**Algorithm S1** Gillespie Sampling

**input** Model $Q_\theta$, start sequence $x$, branch length $t$
**output** Samples $y \sim P(\cdot \mid x, t)$
1: $t' \leftarrow 0$
2: **while** $t' < t$ **do**
3:    $\lambda_x \leftarrow -\sum_{\ell=1}^{L} Q_\theta(x)_{x_\ell, x_\ell}$
4:    $\tau \sim \text{Exp}(\lambda_x)$
5:    **if** $t' + \tau > t$ **then**
6:      Return $y \leftarrow x$
7:    **end if**
8:    Sample $(\ell^*, a^*)$ with
      $P(\ell, a) = (Q_\theta(x)_\ell)_{x_\ell, a}/\lambda_x$
9:    $x_{\ell^*} \leftarrow a^*$
10:   $t' \leftarrow t' + \tau$
11: **end while**
12: Return $y \leftarrow x$

---

**Algorithm S2** Guided Gillespie Sampling

**input** Model $Q_\theta$, start sequence $x$, time $t$, predictor $\mu_\theta, \sigma_\theta$, scale $\gamma$
**output** Samples $y \sim P(\cdot \mid x, t, z)$
1: $t' \leftarrow 0$
2: **while** $t' < t$ **do**
3:    $g \leftarrow \nabla_x \mu_\theta(x)$
4:    $\tilde{Q}_{x,y} \leftarrow Q_{x,y} \cdot [2\Phi(g^\top(y-x)/\sigma_\theta(x))]^\gamma$
5:    $\lambda_x \leftarrow \sum_{y \neq x} \tilde{Q}_{x,y}$
6:    $\tau \sim \text{Exp}(\lambda_x)$
7:    **if** $t' + \tau > t$ **then**
8:      Return $y \leftarrow x$
9:    **end if**
10:   Sample $(\ell^*, a^*)$ with $P(\ell, a) = (\tilde{Q}_\theta(x)_\ell)_{x_\ell, a}/\lambda_x$
11:   $x_{\ell^*} \leftarrow a^*$
12:   $t' \leftarrow t' + \tau$
13: **end while**
14: Return $y \leftarrow x$

---

**A.3. Comparison of CoSiNE against DASM+Thrifty**

During training, DASM (Matsen et al., 2026) explicitly factorizes the affinity maturation process into the product of a somatic hypermutation (SHM) process $q(y \mid x, t)$, with a selection function $F(y)$. Inferring both $q$ and $F$ simultaneously from data is not possible since there is now an extra degree of freedom. Instead, DASM utilizes a frozen Thrifty SHM model $q_\phi$, trained on out-of-frame transitions, and proposes to learn $F_\theta(y)$ via MLE of the observed transitions.

$$p(y \mid x) = \prod_{\ell=1}^{L} p(y_\ell \mid x, t) = \prod_{\ell=1}^{L} q(y_\ell \mid x, t) F_\theta(y_\ell \mid x) \quad \text{s.t.}$$

$$p(x_\ell \mid x, t) = 1 - \sum_{a \in \mathcal{A}} q(a \mid x, t) F_\theta(a \mid x)$$

The second constraint essentially subsumes the normalizing constant into the probability of no-transition. The authors clamp the sum on the right-hand side of the constraint to be less than 1 in order to maintain a valid probability distribution.

Instead of using this formulation, which requires manual clamping of the selection scores, CoSiNE infers $F_\theta(y \mid x)$ at inference time using a log-likelihood ratio between our pre-trained affinity maturation and somatic hypermutation models (Equation (5)).

**A.4. Comparison of CoSiNE against Discrete Flow Matching**

While CoSiNE and Discrete Flow Matching (DFM) (Gat et al., 2024; Campbell et al., 2024) both rely on continuous-time Markov chains, they serve fundamentally different objectives. DFM is a dynamical generative framework that maps a tractable prior (e.g., uniform noise) to a target data distribution. This transformation occurs over an artificial algorithmic time horizon, typically $\tau \in [0, 1]$, where intermediate states act as a computational mechanism for exact sampling rather than representing a physical temporal process. In contrast, CoSiNE is a phylogenetic model designed to learn biological transition kernels. It explicitly models the evolutionary trajectory from an ancestral sequence to a descendant sequence over real, unbounded biological time ($t > 0$).

Although their core objectives differ, connecting these frameworks offers practical directions for future work. For example, DFM could be used to learn expressive stationary priors to mathematically constrain the evolutionary trajectories modeled by CoSiNE, or flow-matching objectives could be adapted to operate over the unnormalized biological timescales required for phylogenetic inference.

# B. Experimental Details

## B.1. Synthetic Experiments on Single Codons

To empirically validate Proposition 4.1 and Lemma 4.2, we utilized a computationally tractable state space that allows for manipulation of the full rate matrix. Specifically, we modeled short DNA sequences with length $L = 3$ over the vocabulary $\mathcal{D} = \{A, G, C, T\}$, thus obtaining a state space that corresponds exactly to the 64 standard codons. To study processes with different levels of epistasis, we constructed ground truth rate matrices using linear interpolation between two extremes:

$$\mathbf{Q}_{\text{true}} = (1 - \varepsilon)\mathbf{Q}_{\text{factorized}} + \varepsilon\mathbf{Q}_{\text{state-dep}}$$

where $\varepsilon \in [0, 1]$ is the epistasis strength parameter. For $\mathbf{Q}_{\text{factorized}}$ (no epistasis), we sampled site-independent base rates: for each site $\ell \in \{1, 2, 3\}$, we drew a $4 \times 4$ rate matrix with off-diagonal entries from Uniform$(0, 2)$ and diagonal entries set to the negative row sum. The global $64 \times 64$ matrix $\mathbf{Q}_{\text{factorized}}$ assigns rate $r_{\ell,a,b}$ to transitions differing only at site $\ell$ with substitution $a \to b$. For $\mathbf{Q}_{\text{state-dep}}$ (maximum epistasis), each of the $64 \times 9 = 576$ transitions with a Hamming distance equal to 1 receives an independent rate drawn from Uniform$(0, 2)$. At $\varepsilon = 0$, rates are perfectly factorizable; at $\varepsilon = 1$, every transition is state-dependent.

For each $\mathbf{Q}_{\text{true}}$ that we drew in this way, we generated 2.5M training samples by drawing branch lengths $b \sim \text{Exp}(\lambda = 0.5)$, sampling start states $x$ uniformly, and sampling end states from $P(\cdot \mid x, t) = \exp(t\mathbf{Q}_{\text{true}})_{x,\cdot}$. We compared three estimators: (1) **Full MLE**, which directly parameterizes all 576 transition rates and optimizes via gradient descent on the exact likelihood; (2) **Factorized**, a neural model that outputs context-dependent site-level $4 \times 4$ rate matrices and assumes site-independence during training; and (3) **Factorized SNR**, which uses the same architecture as (2) but applies SNR weighting (Section C.1). All models were trained for up to 1000 epochs with Adam (lr $= 0.01$ for factorized models, lr $= 0.1$ for Full MLE) and early stopping (patience=50). We tested $\varepsilon \in \{0, 0.25, 0.5, 0.75, 1.0\}$ with 3 replicates per level, measuring estimation error as the relative difference in Frobenius norm of the estimated and ground truth rate matrices: $\|\mathbf{Q}_{\text{est}} - \mathbf{Q}_{\text{true}}\|_F / \|\mathbf{Q}_{\text{true}}\|_F$.

Using the trained factorized models, we compare Gillespie sampling (Algorithm S1) against per-site matrix exponentiation (Equation (2)). To evaluate each sampling method without sampling noise, we compute exact transition probability distributions analytically. For Gillespie, we achieve this by first reconstructing the full $64 \times 64$ estimated rate matrix by querying the model at all 64 states, then computing the transition probability matrix. We compare both methods for the factorized and SNR-weighted models (4 curves total) by measuring KL divergence to the ground truth transition probability across 30 log-spaced branch lengths from $t = 0.01$ to $t = 10.0$, averaged uniformly over all 64 starting states.

## B.2. Calculating the Categorical Jacobian via Perturbation

To quantify the epistatic interactions learned by COSINE, we approximate the Jacobian by exhaustively computing all single point mutations. This procedure is described in Algorithm S3 with an example output sensitivity matrix depicted in Figure 3.

---

**Algorithm S3** Categorical Jacobian Computation

---

1: **Input:** Antibody sequence $x$ of length $L$, Vocabulary $\mathcal{A}$ ($|\mathcal{A}| = 20$)
2: **Output:** Sensitivity Matrix $\mathbf{S} \in \mathbb{R}^{L \times L}$
3: Initialize $\mathbf{S} \leftarrow \mathbf{0}_{L \times L}$
4: Compute wildtype rate matrices: $Q = Q_\theta(x)$
5: **for** $i = 1$ **to** $L$ **do**
6:     **for** $a \in \mathcal{A}$ **such that** $a \neq \mathbf{x}_i$ **do**
7:         $x' \leftarrow x$
8:         $x'_i \leftarrow a$ {Mutate residue at position $i$ to $a$}
9:         $Q' \leftarrow Q_\theta(x')$
10:         **for** $j = 1$ **to** $L$ **do**
11:             $\mathbf{S}_{i,j} \leftarrow \mathbf{S}_{i,j} + \|\mathbf{Q}_j - \mathbf{Q}'_j\|_F$    {Calculate shift in output at position $j$}
12:         **end for**
13:     **end for**
14:     $\mathbf{S}_{i,:} \leftarrow \mathbf{S}_{i,:} / (|\mathcal{A}| - 1)$    {Average over all possible mutations}
15: **end for**
16: **Return S**

---

## B.3. Variant Effect Prediction Details

### B.3.1. DMS ASSAYS AND BASELINE MODELS

The results shown in Table 1 came from four DMS assays from the FLAb2 benchmark (Chungyoun & Gray, 2025). We provide more details on these datasets in Table S1. Following Matsen et al. (2026), we obtain the Koenig and Shanehsazzadeh (Shaneh.) datasets from commit 67738ee (April 17, 2024) of the FLAb Github repository and evaluate all models on these assays. Note that a later commit to the repository modified these assays, so the results we report may differ from those reported by Kim et al. (2026), who use the updated version. We use a single fixed version across all baselines to ensure a controlled comparison. The Adams dataset is taken from commit 3453aeb (September 1, 2025). The Adams dataset contains multiple fitness measurements per mutant sequence, so we correlate antibody model predictions with the average fitness per mutant during VEP evaluation.

An additional step we must take when calculating the COSINE selection score for these assays is determining the underlying nucleotide sequence for the wildtype antibody so that we can calculate the transition likelihood under an SHM model (see Section B.3.2 for more details). For the Koenig and Shaneh. datasets, we use `IMGT/DomainGapAlign` to map the wildtype amino acid sequence to the closest germline V and J genes (Ehrenmann et al., 2010). We then obtain the V- and J-segment nucleotide sequences from HG38 and backtranslate remaining mismatches and junction regions to the codon with the highest frequency in the human genome ((Nakamura et al., 2000)). The Adams dataset uses mouse antibody 4-4-20 scFv as its wildtype, and its nucleotide sequence was obtained from Addgene plasmid pCT302 (Midelfort et al., 2004).

*Table S1.* **Overview of Deep Mutational Scanning (DMS) Datasets.** *Avg. Subs* denotes the average number of amino acid substitutions per mutant relative to wildtype.

| Original Source | Assay Type | Dataset Name | Mutated Chain | Chain Length | | Num. Seqs | Avg. Subs |
| | | | | Heavy | Light | | |
| --- | --- | --- | --- | --- | --- | --- | --- |
| Koenig et al. (2017) | Expression | Koenig Expression (H) | Heavy | 120 | 108 | 2261 | 1 |
| | | Koenig Expression (L) | Light | 120 | 108 | 2014 | 1 |
| | Binding (VEGF) | Koenig Binding (H) | Heavy | 120 | 108 | 2261 | 1 |
| | | Koenig Binding (L) | Light | 120 | 108 | 2014 | 1 |
| Adams et al. (2016) | Expression | Adams | Heavy | 117 | 112 | 2803 | 1.98 |
| Shanehsazzadeh et al. (2023) | Binding (HER2) | Shaneh. (119) | Heavy | 119 | 107 | 184 | 5.22 |
| | | Shaneh. (120) | Heavy | 120 | 107 | 201 | 5.97 |

We evaluate zero-shot VEP with six other baseline models, which can be divided into three categories based on their architectures and how they are evaluated.

1. Masked language models (**AbLang-2, ESM-2 150M, ESM-2 650M**) are evaluated via pseudo-perplexity in accordance with the FLAb2 benchmark.

2. Autoregressive models (**ProGen2 Small, ProGen2 Medium**) are evaluated via perplexity, also in accordance with the FLAb2 benchmark.

3. The **DASM** model is evaluated by summing its log selection factors as described in Matsen et al. (2026).

4. The **PRISM** model is also evaluated via pseudo-perplexity. Following the protocol in Kim et al. (2026), we use the germline scoring mode for expression assays and the non-germline mode for binding.

### B.3.2. CALCULATING SEQUENCE LIKELIHOODS WITH THRIFTY SHM MODEL

To estimate sequence transition likelihoods under SHM, we use `ThriftyHumV0.2-59-hc-tangshm`, which provides per-codon transition probabilities with a multihit correction described in Matsen et al. (2026). Since the Thrifty model operates in the state space of codons (64 states) and COSINE operates in the state space of amino acids (20 states), we must sum over all possible codons that could code for the observed alternate allele when calculating sequence likelihoods.

More formally, let $x_\ell$ denote the amino acid identity at position $\ell$ in the wildtype sequence for a DMS assay and $c(x_\ell)$ denote its underlying nucleotide codon (see Section B.3.1 for details on how this is determined). Let $y_\ell$ denote the corresponding

amino acid in the mutant sequence $y$ and $C(y_\ell)$ denote the set of possible codons that could code for $y_\ell$. We calculate the likelihood of sequence $y$ under Thrifty as

$$q(y \mid x, t) = \prod_{\ell}^{L} q(y_\ell \mid x_\ell, t) \quad \text{where}$$

$$q(y_\ell \mid x_\ell, t) = \begin{cases} q(c(x_\ell) \mid c(x_\ell), t) & \text{if } y_\ell = x_\ell \\ \sum_{c \in C(y_\ell)} q(c \mid c(x_\ell), t) & \text{if } y_\ell \neq x_\ell. \end{cases}$$

### B.4. Guided Affinity Maturation from Naive Antibodies

We randomly sampled naive antibodies from a subset of the OAS database (Olsen et al., 2022a) containing only heavy chain sequences with the IgM isotype from human donors. To guide and score the sampled sequences, we utilized the SARS-CoV-1 and SARS-CoV-2 neutralization predictors from Jin et al., with weights downloaded from `https://github.com/wengong-jin/RefineGNN`. These oracles are only trained on heavy chain sequences. To parameterize the variance $\sigma_{\theta_z}$, we used MC dropout (Gal & Ghahramani, 2015) with 10 fixed masks. For the numerator in Equation (9), we set the model to evaluation mode and computed the gradient of the output $\mu_{\theta_z}(x)$ with respect to the one-hot input sequence $x$. Known binders were curated from the CoV-AbDab database. We set the naive antibody at the root of the clonal tree in Figure S1 and recursively sampled down its nodes in breadth first order. We repeated this procedure 5 times for each guidance setting and collected the leaf sequences for comparison. The selected tree has 13 leaves, so a non-repetitive sampler would produce 65 unique leaf sequences.

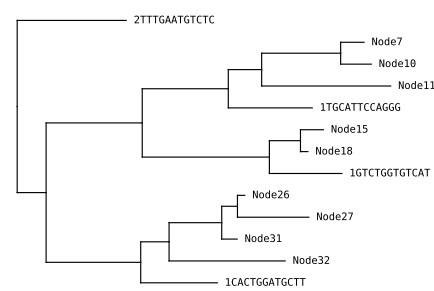

*Figure S1.* **Clonal Tree used for Guided Affinity Maturation Experiment in Section 5.4**

### B.5. Guided Optimization of Antibody CDRs

All methods optimize a SARS-CoV-1 binder from the CovAbDab database, with CDR positions identified via the IMGT numbering scheme (Ehrenmann et al., 2010). The Greedy baseline (described below) is exempt from the oracle-call budget and serves only as an upper bound on achievable affinity. We compare COSINE against the following baselines:

**Genetic Algorithm (GA).** We evolved a population of 1,000 sequences over 9 generations. In each generation, the top 50% of sequences were selected based on fitness, and the remaining 50% were regenerated by applying random mutations to the selected parents. To respect the mutation budget without inefficient rejection sampling, we restrict changes to mutated positions if the budget of 5 mutations has been met. This allows for lateral moves or reversions while satisfying constraints.

**Product of Experts (ESM-2 and AbLang-1).** Following Gordon et al., we sampled variants using protein language models with a product of experts (PoE) strategy to incorporate oracle guidance. This method uses Gibbs sampling from a joint distribution $P(x) \propto P_{\text{MLM}}(x) \cdot \exp(\lambda F(x))$, where $P_{\text{MLM}}$ is a masked language model prior. We evaluated both a general protein language model (ESM-2 150M) and an antibody-specific model (AbLang-1). We used AbLang-1 instead of AbLang-2 because the latter strictly requires paired antibody sequences instead of a single VH chain. We set the guidance strength to $\lambda = 50.0$. To respect the computational budget of $\leq 5$ oracle calls per sample, we implemented a caching approximation. Before sampling, we pre-computed the fitness effects of all possible single mutations in CDRs. During the Gibbs sampling process, oracle scores were retrieved from this additive cache rather than re-evaluated using the mutated sequence. This approximates the fitness landscape as locally additive.

**Greedy Search.** As a performance upper bound, we implemented a stochastic hill-climbing strategy. At each step, the algorithm evaluates all possible single mutations ($L \times 19$ variants), selects the top $K = 15$ candidates by fitness gain, and samples the next step using a softmax over their fitness improvements ($\Delta F/T$, with $T = 1.0$). This process repeats for up to 5 steps. While effective, this method requires orders of magnitude more compute ($\sim$2700 oracle calls per sequence) and serves only as a reference for the maximum achievable affinity under the given constraints.

**Evaluation Metrics.** Performance was evaluated across three axes: **Fitness**, measured by the mean and maximum improvement in predicted binding affinity; **Diversity**, quantified by the average pairwise distance within the generated samples; and **Naturalness**, measured using the OASis score.

# C. Supplementary Results

## C.1. Estimation Error of $Q_\theta(x)_\ell$ via SNR-Weighted MLE

Under the transition likelihood of the COSINE model in Equation (2), maximum likelihood inference of $Q_\theta(x)_\ell$ is performed by minimizing the negative log-likelihood of the observed evolutionary transitions

$$\mathcal{L}(\theta) = - \sum_{\tau=(x,y,t)} \sum_{\ell=1}^{L} \log \exp(tQ_\theta(x)_\ell)_{x_\ell, y_\ell}$$

Unfortunately, directly training with this objective will generally fail to satisfy the assumption in Proposition 4.1, which relies on $(Q_\theta(x)_\ell)_{x_\ell,\cdot}$ approximating the instantaneous non-zero rates in $\mathbf{Q}_{x,\cdot}$. Indeed, on branches with large $t$, this factorized objective encourages $Q_\theta(x)_\ell$ to learn *effective* rates that account for unobserved epistatic interactions and intermediate states between $x$ and $y$. This results in a time scale dependent bias where the inferred rates for long branches may diverge from $\mathbf{Q}$.

To mitigate this issue, we sought to leverage the insight that the signal-to-noise (SNR) ratio of our first-order approximation scales as $O(1/t)$. We therefore proposed a SNR-weighted loss function:

$$\mathcal{L}_{SNR}(\theta) = \sum_{\tau=(x,y,t)} \frac{1}{\delta + t} \mathcal{L}(\theta; x, y, t)$$

where $\delta \in [0,1]$ is a hyper-parameter that controls the magnitude of the weighting. In the synthetic codon experiments (detailed in Section B.1), we were able to show that in a data-rich setting, SNR weighting indeed reduces the estimation error compared to an unweighted MLE objective, especially at high epistasis levels for the ground truth rate matrix (Figure S2). In addition, we found that setting $\delta = 1 - \varepsilon$ yielded the strongest results, which is intuitive since re-weighting is meant to compensate for epistatic effects in the underlying process.

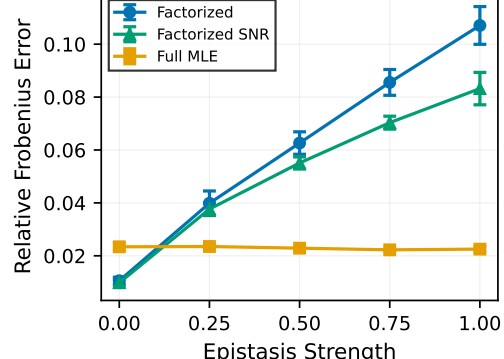

*Figure S2.* **As $\varepsilon$ increases, SNR-weighting (green) reduces the relative Frobenius norm error between the estimated and true rate matrices compared to unweighted MLE (blue), while the Full MLE (yellow) is constant.**

Motivated by this result, we trained COSINE on the full clonal dataset using this SNR-weighted objective at different values of $\delta \in \{0.0, 0.5, 1.0\}$. However, we could not identify improvements for these models in comparison to a COSINE model trained with the un-weighted loss. We hypothesize that this is due to a relatively data-poor setting for the antibody environment, given that our state space is much larger than the synthetic environment and we have less training data.

## C.2. Sampling Error in Synthetic Codon Experiment

Using the synthetic codon environment detailed in Section B.1, we sought to quantify the difference between the sampling distributions of COSINE and the true transition probability distribution of the underlying process. For each level of epistasis $\varepsilon \in \{0.0, 0.25, 0.5, 0.75, 1.0\}$, we generated a ground truth rate matrix and trained a COSINE model on simulated transitions (see Section B.1). We evaluate performance by measuring the KL divergence between the ground truth transition distribution $P(y \mid x, t)$ and the distributions induced by either Gillespie sampling (Algorithm S1) or the per-site matrix exponentiation approach (Equation (2)).

We found that Gillespie sampling is superior at all epistasis levels and branch lengths. As expected, both sampling methods perform comparably when $\varepsilon = 0$, since the ground truth rate matrix is also site-independent (Figure S3a). However, at $\varepsilon = 1.0$, notice that the KL divergence of the matrix exponential scales quadratically in $t$ before plateauing at a high error, indicating a failure to capture the correct stationary distribution (Figure S3e). In contrast, Gillespie sampling maintains consistently lower divergence and recovers the correct stationary distribution. This is an important result since it empirically validates Proposition 4.1 and supports the claim in Lemma 4.2. As $\varepsilon$ increases, we notice that the advantage of Gillespie sampling over the factorized matrix exponential approach also becomes more significant.

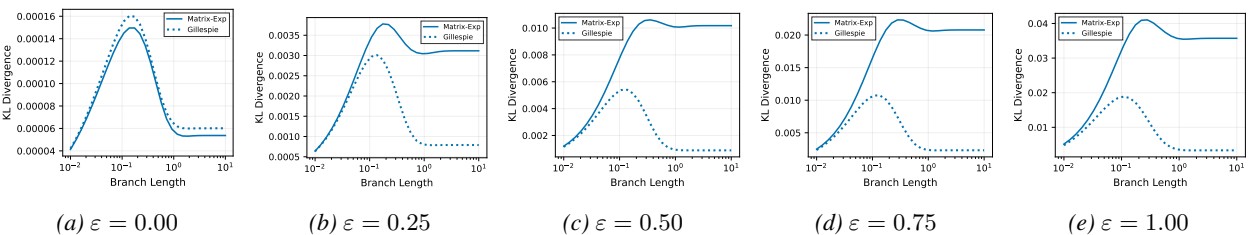

*(a)* $\varepsilon = 0.00$    *(b)* $\varepsilon = 0.25$    *(c)* $\varepsilon = 0.50$    *(d)* $\varepsilon = 0.75$    *(e)* $\varepsilon = 1.00$

*Figure S3.* **Gillespie samples are more consistent with the true transition probability distribution than samples from the factorized COSINE likelihood at all levels of epistasis.** The difference increases smoothly with the epistasis strength $\varepsilon$ in the underlying rate matrix. Ground truth matrix generation and estimation protocols are in Section B.1.

### C.3. Sampling Error in Real Antibody Experiment

Although we cannot obtain the transition likelihood for Gillespie samples in the large antibody sequence state space, we sought to validate the synthetic codon results (Section C.2) in the real antibody environment using auxiliary metrics. We selected all clonal trees in the test split with $\geq 4$ leaves and sampled new sequences for each tree using the model trained in Section 5.1, conditional on the root sequence. For each leaf node in each tree, we collected the corresponding sampled sequence from both Gillespie and factorized matrix exponential approaches and compared their hamming distance to the real leaf sequence at that node. Assuming that a lower hamming distance to the real leaf sequence indicates lower sampling error, we observe in Figure S4a that leaves simulated with Gillespie sampling are closer to the real leaf sequence in 52.0% of cases, whereas the opposite is true only 38.8% of the time. Next, we computed the hamming distance from the root sequence to every real and simulated leaf sequence. Under the assumption that this distance should be somewhat consistent for corresponding real and simulated leaves, we scatterplot their correlation in Figure S4b for Gillespie samples and Figure S4c for factorized matrix exponential samples. Once again, we find that Gillespie has superior performance, achieving a Pearson correlation of $0.64$ while the per-site matrix exponentiation approach only obtains $0.56$. Altogether, these results further consolidate our results in the synthetic codon environment and strongly suggest that Gillespie sampling is a principled and superior sampling approach for COSINE.

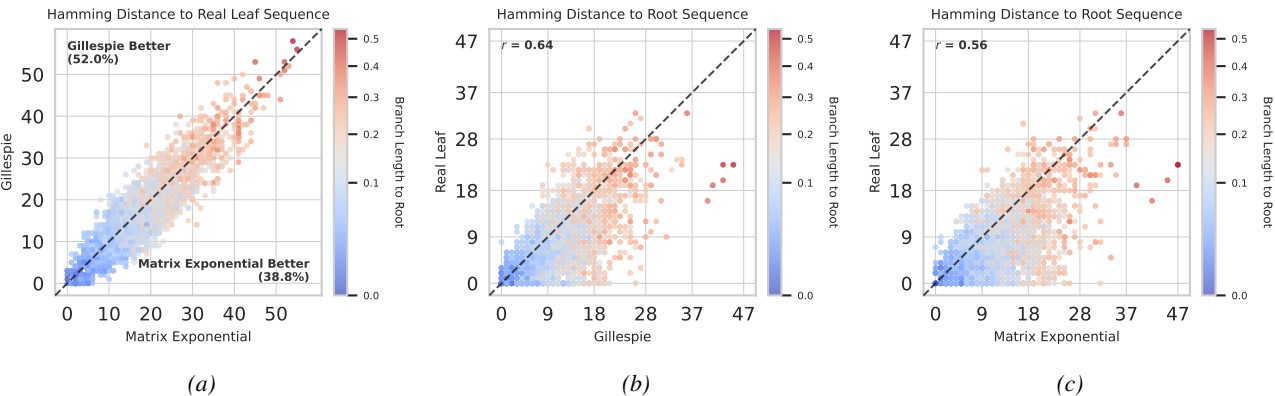

*(a)*    *(b)*    *(c)*

*Figure S4.* **Gillespie sampling exhibits lower sampling error and better preserves evolutionary distances on real antibody clonal families.** a) Comparison of sampling fidelity: Gillespie samples are closer (in Hamming distance) to the true held-out leaf sequence in 52.0% of cases, compared to 38.8% for the factorized matrix exponential. b, c) Correlation between the root-to-leaf Hamming distances of real versus simulated leaf sequences. Gillespie sampling (b) achieves a higher Pearson correlation ($r = 0.62$) with the observed evolutionary distances than the factorized matrix exponential (c) approach ($r = 0.53$).

### C.4. Per-Site Entropy Increases with Branch Length

From a randomly selected parent sequence in the test set, we calculated the per-site entropy of the COSINE transition likelihood in Equation (2) at different branch lengths (Figure S5). As expected, we found that the entropy increases with the branch length, indicating a larger number of expected mutations. Furthermore, we found that the entropy in CDR regions (red) tends to be higher than the entropy in framework regions (grey).

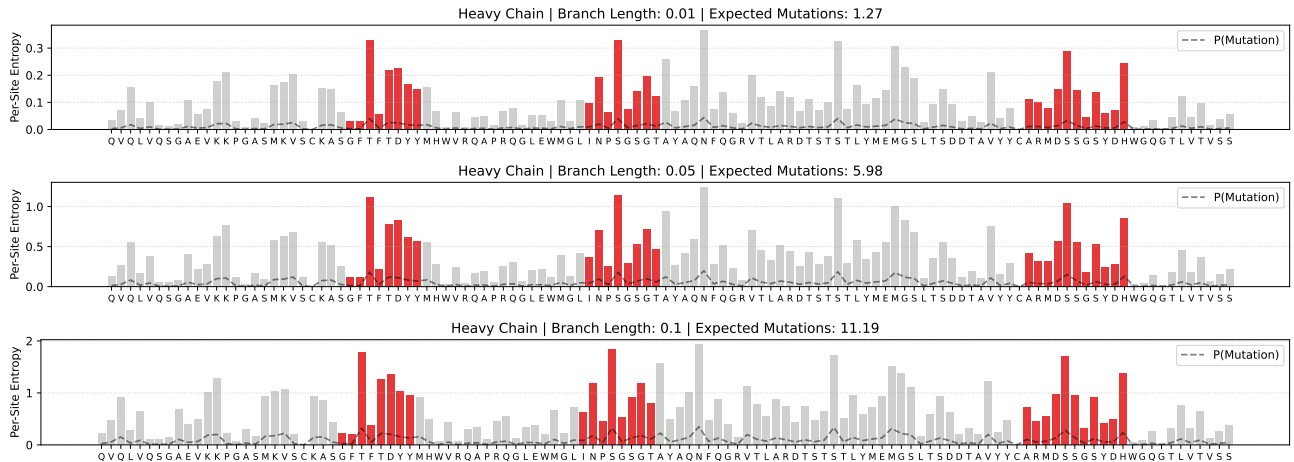

*Figure S5.* **Per-site entropy across increasing branch lengths.** Entropy profiles for branch lengths 0.01, 0.05, and 0.1 (top to bottom). Red bars denote CDRs. Entropy scales with branch length, showing higher mutational variability in CDRs.

### C.5. Ablation of Context Dependent Rate Estimation

We investigate the importance of context-dependence by comparing CoSiNE against a classical context-independent substitution model: a single 20×20 exchangeability matrix fit using the WAG likelihood form. Following the original WAG paper, we parameterized the rate matrix as time-reversible: $Q = S_\theta \cdot \mathrm{diag}(\pi)$. We set the equilibrium frequencies ($\pi$) to the empirical amino acid frequencies in the training data and optimized the exchangeability matrix $S_\theta$ via maximum likelihood using the L-BFGS optimizer, monitoring validation NLL for early-stopping. We estimate these parameters via MLE on the same training transitions used for CoSiNE, ensuring a fair comparison.

*Table S2.* **Comparison of WAG and CoSiNE per-token NLL on test transitions by branch length.**

| Branch Length | WAG NLL | CoSiNE NLL |
|---|---|---|
| $[0, 0.01)$ | 0.0717 | **0.0663** |
| $[0.01, 0.05)$ | 0.171 | **0.147** |
| $[0.05, 0.15)$ | 0.454 | **0.378** |
| $[0.15, \infty)$ | 0.836 | **0.676** |

On our test set, CoSiNE achieves significantly lower NLL per token across all branch lengths, with the gap widening for longer branches, suggesting that sequence-context conditioning allows CoSiNE to capture epistatic interactions that accrue over long evolutionary distances.

### C.6. Evolution of General Protein Families with CoSiNE

To evaluate robustness to both longer branches and noisier tree reconstructions, we trained CoSiNE on the TrRosetta (Yang et al., 2019) dataset (∼15k general protein family MSAs). These families span far greater evolutionary distances than antibody clonal trees, and their phylogenies are reconstructed from sequence homology rather than direct lineage tracing, making them inherently noisier. We compare against two classical substitution models, using the WAG and LG+G4 likelihood forms to fit new exchangeability matrices from scratch on the same training data. CoSiNE achieves lower test per-token NLL than both baselines across all splits, indicating that context-dependent rate modeling reduces misspecification even under these more challenging conditions.

*Table S3.* **Comparison of WAG, LG+G4, and CoSiNE per token NLL on held out transitions from TrRosetta.**

| Model | Val NLL | Test NLL |
|---|---|---|
| WAG | 1.170 | 1.200 |
| LG+G4 | 1.139 | 1.168 |
| CoSiNE | **1.005** | **1.027** |

### C.7. Log-Likelihood versus Selection Score for DMS Performance

Figure S6 illustrates the effect of our correction method on the Koenig Light Chain expression dataset. In the left plot, there is a clear separation in the data according to the number of nucleotide edits between the wildtype and mutant sequences. This indicates that the model has learned that for a fixed amount of elapsed time, transitions with fewer nucleotide edits are more probable than transitions with more nucleotide edits. This makes sense considering our chosen branch length is somewhat short ($t = 0.2$) and the model is trained to maximize the likelihood of observed transitions. However, if we

naively correlate the model's predicted likelihood for the sequence with selection, we introduce the bias that mutations with few nucleotide edits are higher fitness than mutations with more nucleotide edits, which is clearly erroneous. In the plot on the right, we can see that taking the ratio with the likelihood under the Thrifty model removes this edit distance bias as indicated by the fact that the point clouds corresponding to each edit distance now roughly occupy the same space.

The selection score correction does more than just correct for this edit distance bias. For example, certain nucleotide substitutions are more likely to occur than others and without accounting for this variation, those biases will be interpreted as improvements in fitness. In Figure S6 we see that even among mutants with the same nucleotide edit distance, the correlation improves when adding the SHM correction.

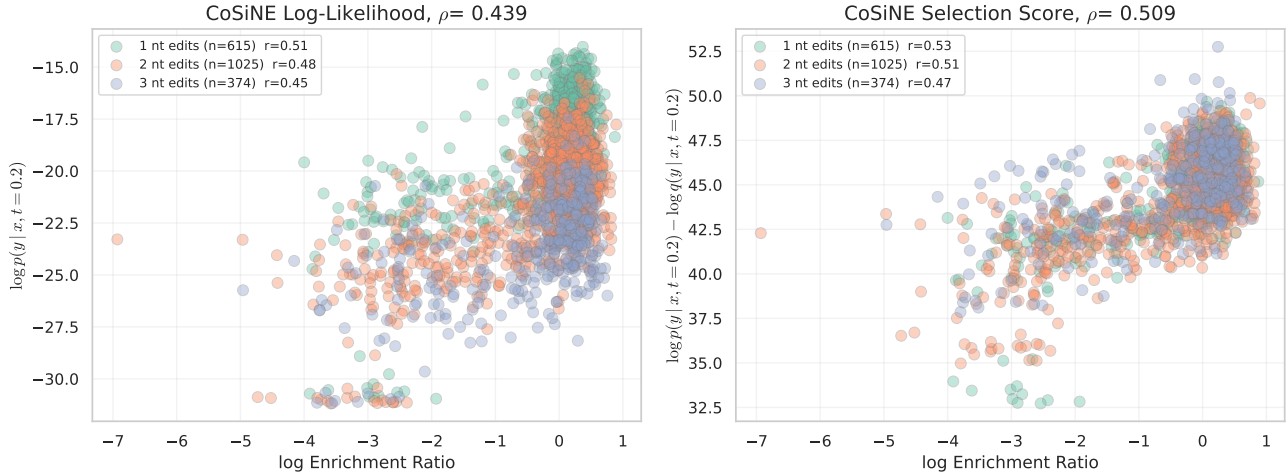

*Figure S6.* **Analysis of CoSiNE scoring methods against log Enrichment Ratios for the Koenig DMS assay measuring expression (Light Chain). Left**: Log-Likelihood performance ($\rho = 0.439$). **Right**: Selection Score performance ($\rho = 0.509$). Points are colored by the edit distance from wildtype codon to mutant codon. We assume the mutant codon is the one which requires the smallest number of nucleotide edits.

## C.8. Additional VEP Experiments

To strengthen our claims of strong VEP performance, we evaluate CoSiNE on two additional VEP datasets from (Petersen et al., 2024). We chose these assays because they use distinct antibodies, antigens, and experimental technologies (MAGMA-seq) from those in Table 1. As seen in Table S4, CoSiNE outperforms all baselines, consistent with previous results.

In Table S5 we evaluate CoSiNE on the same datasets as Section 5.3 when conditioning on different amounts of context. Surprisingly, conditioning on just a single chain results in a higher correlation on four out of seven datasets. However, when paired conditioning outperforms, it does so by an average Spearman correlation of **0.121**, compared to just **0.006** when single chain conditioning outperforms. This suggests that inter-chain interactions are crucial for modeling fitness but only under certain contexts. What differentiates these contexts is a direction for future work. It's also worth noting that the majority of our training data consists of unpaired single-chain sequences, which likely limits the model's ability to fully leverage paired context. We expect that as more paired heavy-light sequencing data becomes available, the benefits of cross-chain conditioning will become more consistent.

In Table S6, we report results from the same experiment as Table 1, but we use Pearson correlation instead of Spearman.

*Table S4.* **VEP Performance of CoSiNE against baselines on Petersen binding datasets (Spearman).**

| Model | Petersen (319-345) | Petersen (222-1C06) |
|---|---|---|
| AbLang-2 | 0.199 | 0.060 |
| DASM | 0.395 | 0.286 |
| PRISM | 0.312 | −0.073 |
| ESM2-150M | 0.250 | −0.139 |
| ESM2-650M | 0.278 | 0.013 |
| ProGen2-Small | 0.329 | 0.024 |
| ProGen2-Medium | 0.294 | 0.036 |
| CoSiNE ($t = 0.2$) | **0.504** | **0.328** |

*Table S5.* **Comparison of CoSINE on zero-shot VEP with both the heavy and light chains provided as context (Paired) versus just the chain with the mutations (Single).** Blue indicates datasets with mutations on the heavy chain, and red indicates mutations on the light chain.

| MODEL | EXPRESSION | | | BINDING | | | |
|---|---|---|---|---|---|---|---|
| | KOENIG (H) | KOENIG (L) | ADAMS | KOENIG (H) | KOENIG (L) | SHANEH. (119) | SHANEH. (120) |
| CoSINE-PAIRED | **0.613** | 0.508 | **0.464** | **0.456** | 0.371 | 0.498 | 0.536 |
| CoSINE-SINGLE | 0.545 | **0.509** | 0.234 | 0.390 | **0.375** | **0.504** | **0.549** |

*Table S6.* **Comparison of deep protein models on VEP benchmarks across expression and binding landscapes as measured by Pearson correlation.**

| MODEL | EXPRESSION | | | BINDING | | | |
|---|---|---|---|---|---|---|---|
| | KOENIG (H) | KOENIG (L) | ADAMS | KOENIG (H) | KOENIG (L) | SHANEH. (119) | SHANEH. (120) |
| ABLANG-2 | 0.153 | −0.109 | −0.096 | −0.114 | −0.108 | 0.263 | 0.166 |
| DASM | **0.688** | 0.674 | 0.221 | 0.335 | 0.316 | 0.458 | 0.518 |
| PRISM | 0.055 | 0.145 | 0.243 | −0.001 | 0.000 | 0.346 | 0.251 |
| ESM2-150M | 0.476 | 0.539 | −0.119 | 0.044 | 0.266 | 0.215 | 0.197 |
| ESM2-650M | 0.384 | 0.416 | 0.097 | 0.009 | 0.243 | 0.191 | 0.308 |
| PROGEN2-SMALL | 0.559 | 0.568 | −0.043 | 0.156 | 0.276 | 0.074 | 0.052 |
| PROGEN2-MEDIUM | 0.553 | 0.579 | 0.209 | 0.123 | 0.253 | 0.296 | 0.275 |
| CoSINE (OURS) | 0.687 | **0.696** | **0.409** | **0.367** | **0.345** | **0.502** | **0.521** |

## C.9. Sensitivity of VEP to the Choice of Branch Length

We investigate the robustness of CoSINE to the choice of the branch length hyperparameter, $t$, for variant effect prediction (VEP) tasks. In Figure S7 we evaluate CoSINE across the datasets in Section 5.3 using six branch length values ranging from $t = 0.01$ to $t = 10$, and Table S7 reports the underlying values.

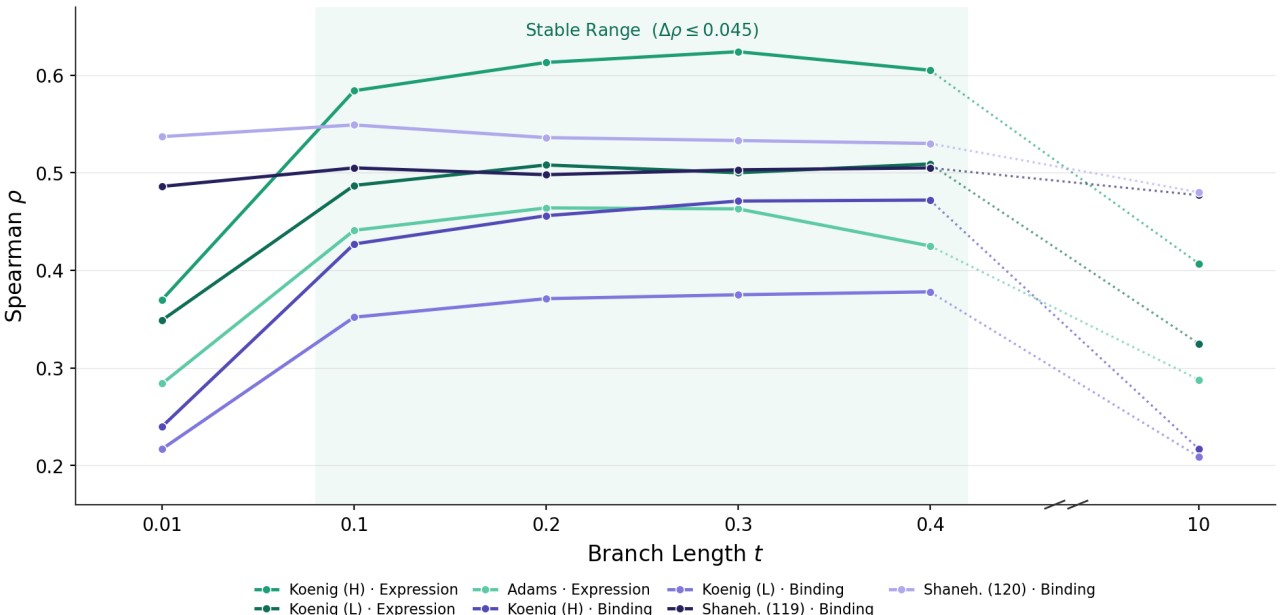

*Figure S7.* **Spearman correlation $\rho$ between the CoSINE selection score and experimental fitness evaluated over branch lengths $t \in \{0.01, 0.1, 0.2, 0.3, 0.4, 10\}$.** Expression assays are shown in green and binding assays are shown in purple.

The selection score derived from CoSINE demonstrates strong reliability across a relatively broad range from $t = 0.1$ to $t = 0.4$, where the maximum difference in correlation ($\Delta\rho$) across settings is only $0.045$. Within this optimal range, CoSINE outperforms all baseline models on at least five of the seven assays and achieves second-best performance on

*Table S7.* **Underlying Spearman correlation values for Figure S7**. CoSiNE results are **bolded** where they tie or exceed all non-CoSiNE baselines (Table 1).

| MODEL | EXPRESSION | | | BINDING | | | |
|---|---|---|---|---|---|---|---|
| | KOENIG (H) | KOENIG (L) | ADAMS | KOENIG (H) | KOENIG (L) | SHANEH. (119) | SHANEH. (120) |
| CoSiNE ($t = 0.01$) | 0.370 | 0.349 | 0.284 | 0.240 | 0.217 | **0.486** | **0.537** |
| CoSiNE ($t = 0.1$) | 0.584 | 0.487 | **0.441** | **0.427** | **0.352** | **0.505** | **0.549** |
| CoSiNE ($t = 0.2$) | **0.613** | 0.508 | **0.464** | **0.456** | **0.371** | **0.498** | **0.536** |
| CoSiNE ($t = 0.3$) | **0.624** | 0.500 | **0.463** | **0.471** | **0.375** | **0.503** | 0.533 |
| CoSiNE ($t = 0.4$) | **0.605** | 0.509 | **0.425** | **0.472** | **0.378** | **0.505** | 0.530 |
| CoSiNE ($t = 10$) | 0.407 | 0.325 | 0.288 | 0.217 | 0.209 | **0.477** | 0.480 |

the remaining two. As expected, performance degrades at the extremes: at very small branch lengths ($t = 0.01$) the transition signal is dominated by noise, while at very large branch lengths ($t = 10$) the quadratic error term (Proposition Proposition 4.1) grows and the factorized likelihood diverges from the true process. Nonetheless, the correlations remain significantly positive.

While performance could theoretically be maximized by calibrating the choice of $t$ per assay, we utilize a fixed value of $t = 0.2$ across all main experiments. This decision prevents unfair fitting to the VEP benchmarks, as real-world applications of zero-shot prediction lack access to ground-truth fitness values for hyperparameter tuning.

### C.10. Ablation of ESM2 Backbone

To determine the source of CoSiNE's performance gain and to assess the specific impact of the pretrained backbone, we performed an ablation study on the ESM2 initialization. We trained a version of CoSiNE entirely from scratch (comprising 8 million parameters with random weight initialization) using the same training dataset. This setup allows us to isolate the predictive power derived from the pretrained ESM2-150 component versus the evolutionary training objective itself on VEP.

*Table S8.* **Ablation of the pretrained ESM2 backbone on VEP benchmarks.** Spearman correlation ($\rho$) is reported across expression and binding datasets. Best performing models are shown in **bold**; second-best are underlined.

| MODEL | EXPRESSION | | | BINDING | | | |
|---|---|---|---|---|---|---|---|
| | KOENIG (H) | KOENIG (L) | ADAMS | KOENIG (H) | KOENIG (L) | SHANEH. (119) | SHANEH. (120) |
| DASM | 0.596 | 0.474 | 0.270 | 0.415 | 0.327 | 0.450 | **0.536** |
| ESM2-650M | 0.326 | 0.429 | 0.124 | 0.063 | 0.265 | 0.227 | 0.360 |
| PROGEN2-SMALL | 0.407 | **0.513** | -0.024 | 0.098 | 0.332 | 0.119 | 0.070 |
| CoSiNE-ESM2 | **0.613** | 0.508 | **0.464** | **0.456** | **0.371** | 0.498 | **0.536** |
| CoSiNE-8M | 0.503 | 0.474 | 0.447 | 0.363 | 0.330 | **0.508** | 0.531 |

As shown in Table S8, the ESM2 backbone improves CoSiNE's selection score correlation with fitness on all but one dataset. However, the from-scratch 8M version still performs exceptionally well: the average difference in correlation ($\Delta\rho$) between the ESM2-initialized version and the 8M version is only $0.041$. This suggests that while the pretrained ESM2 backbone provides a highly beneficial initialization, it is not the primary source of predictive power.

### C.11. Taylor-Series Approximated Guidance versus Exact Guidance

To evaluate the utility of our first-order Taylor series approximation for oracle guidance, we compared the fitness improvements and computational costs of exact guidance versus Taylor series approximated guidance. Using the same SARS-CoV-1 oracle as before (Section B.4), we sampled sequences at $t = \{0.01, 0.05, 0.10\}$ with guidance strength $\gamma = 2.0$. We generated 5 samples from each of 3 randomly selected seed antibody sequences, yielding 15 samples per condition.

As shown in Figure S8 (left), both guidance methods produced similar fitness improvements across all branch lengths. Two-sided $t$-tests confirmed that these differences were not statistically significant at any branch length ($p = 0.571$, $p = 0.918$, and $p = 0.695$), indicating that Taylor approximation does not compromise the quality of *Guided Gillespie* samples.

In contrast, the computational costs differed dramatically between methods (Figure S8, right). Exact guidance requires

evaluating the oracle on all single-amino-acid mutants at each Gillespie step, compared to a single call for TAG guidance. This achieves speedups of 488× (0.01), 928× (0.05), and 916× (0.10) across the three branch lengths.

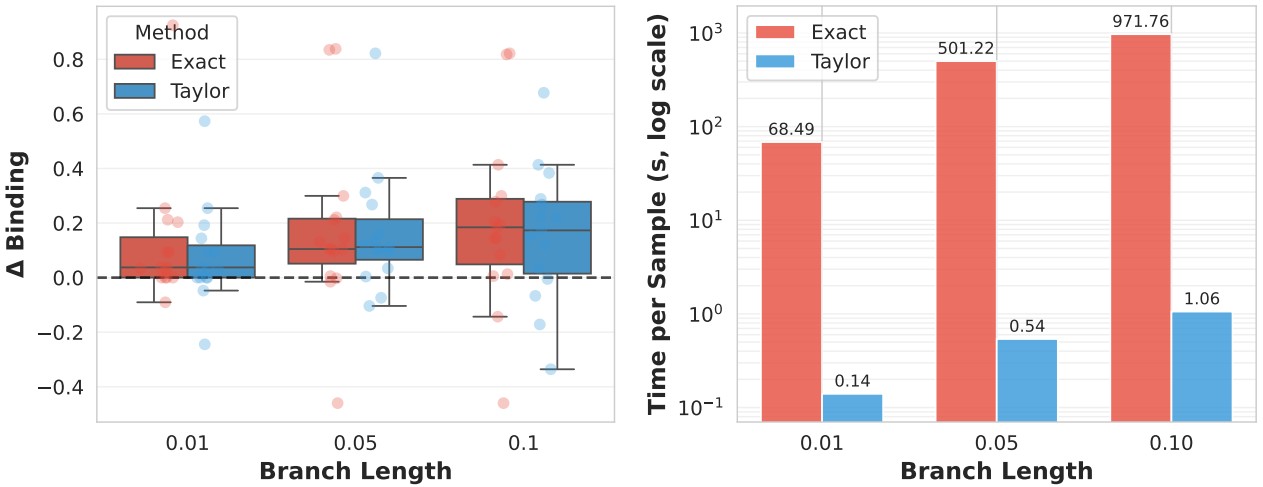

*Figure S8.* **TAG guidance matches exact guidance performance with 500–900× speedup**. (Left) Fitness improvements and (right) runtime comparison across branch lengths. No significant fitness differences ($p > 0.05$), but TAG is orders of magnitude faster.

## C.12. Runtime Analysis of *Guided Gillespie*

In this section, we analyze the computational cost of *Guided Gillespie* sampling, as seen in Equation (9), and its scaling properties with respect to the sequence length $L$ and the number of sampling steps $N$.

Empirically, the runtime scales linearly with both sequence length and the number of sampling steps. Table S9 illustrates these scaling properties. When the number of sampling steps is fixed ($N = 50$), the guided runtime increases linearly with $L$ due to the computational cost of the predictor's forward pass. In contrast, the unguided runtime remains constant across lengths because it makes no calls to the predictor. Conversely, when the sequence length is fixed ($L = 100$), both guided and unguided runtimes scale linearly with $N$.

*Table S9.* **Runtime analysis of Guided vs. Unguided Gillespie sampling. Left:** Runtime scaling by sequence length $L$ with a fixed $N = 50$ sampling steps. **Right:** Runtime scaling by sampling steps $N$ with a fixed sequence length of $L = 100$.

| Sequence Length ($L$) | Guided Runtime (s) | Unguided Runtime (s) | | Sampling Steps ($N$) | Guided Runtime (s) | Unguided Runtime (s) |
|---|---|---|---|---|---|---|
| 50 | 5.16 | 3.54 | | 10 | 1.22 | 0.73 |
| 100 | 5.94 | 3.51 | | 25 | 3.07 | 1.76 |
| 250 | 8.53 | 3.51 | | 50 | 6.14 | 3.50 |
| 500 | 12.20 | 3.65 | | 100 | 12.20 | 6.96 |

## C.13. Guided Affinity Maturation from Additional Naive Antibodies

We provide additional results for the guided affinity maturation sampling experiment using other naive sequences from the OAS database. Notice that in all cases, guidance effectively steers the predicted binding affinity of the generated leaf sequences.

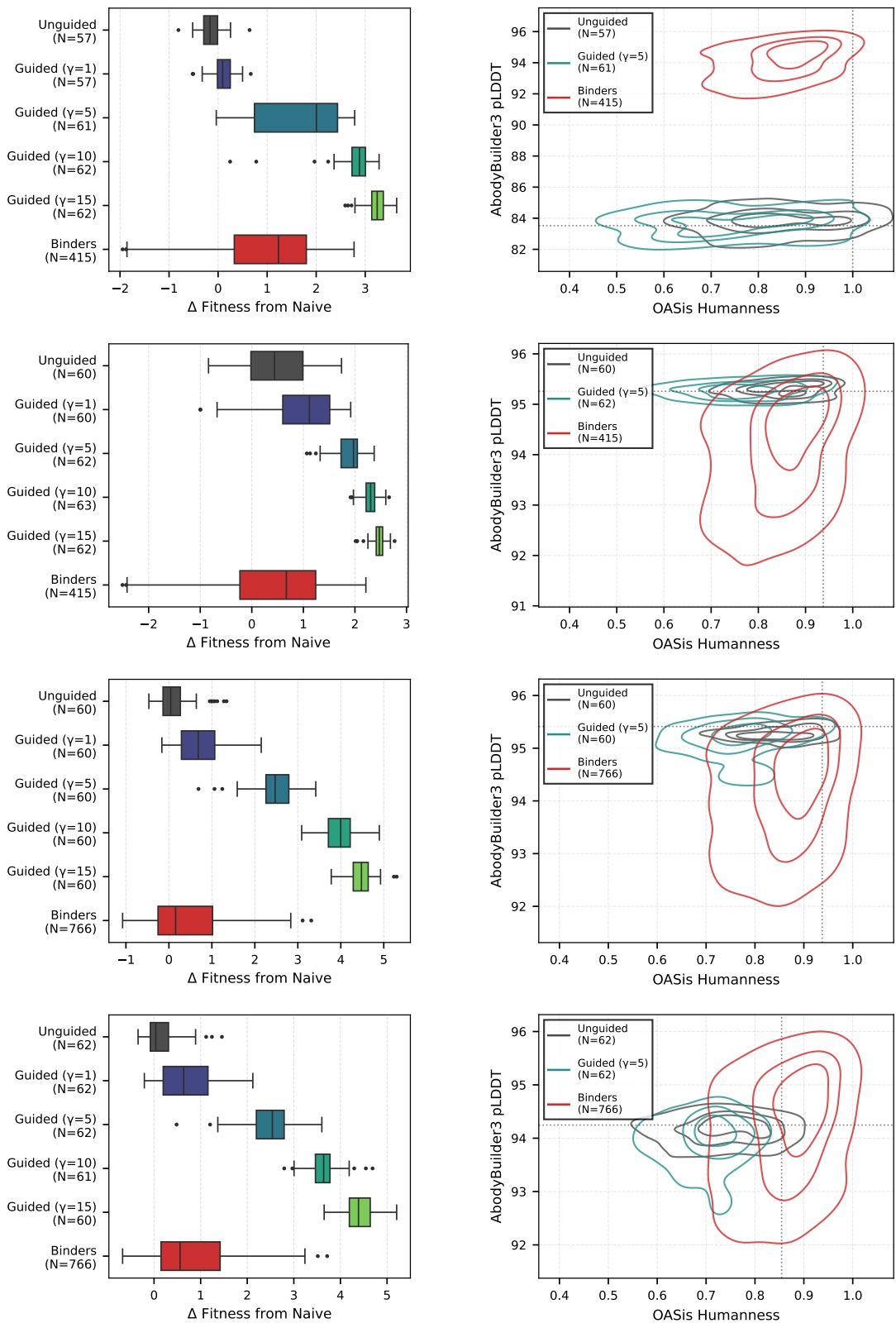

*Figure S9.* **Guided affinity maturation results for additional naive antibody sequences using CoV-1 (top) and CoV-2 (bottom) oracles.** We randomly selected naive IgM heavy sequences from the OAS database and recursively sampled down the tree in Figure S1.

# D. Theoretical Results

## D.1. First-Order Approximation of Sequential Point Mutation Process

**Proposition D.1.** *(Proof of Proposition 4.1)*

*Assume the per-site rate matrices $Q_\theta(x)_\ell$ are parameterized such that*

$$(Q_\theta(x)_\ell)_{x_\ell, y_\ell} = \mathbf{Q}_{x,y}$$

*for all $x, y$ such that Hamming distance $d(x, y) = 1$ and $\ell$ is the unique site where $x$ and $y$ differ. Then, the error between the transition probability vectors is bounded such that*

$$\|P(\cdot \mid x, t) - p_\theta(\cdot \mid x, t)\|_1 \leq (\lambda t)^2 = O(t^2) \tag{10}$$

*where $\lambda$ is the maximum exit rate of $\mathbf{Q}$.*

*Proof.* Notice that we can reframe the transition probabilities from the per-site rate matrices using Kronecker products as follows

$$p_\theta(\cdot \mid x, t) = \left( e^{tQ_\theta(x)_1} \otimes \cdots \otimes e^{tQ_\theta(x)_L} \right)_{x, \cdot}$$

Using the identity $e^{tA} \otimes e^{tB} = e^{t(A \oplus B)}$ we can define a **Kronecker-sum generator matrix**

$$\mathbf{Q}_\theta(x) := Q_\theta(x)_1 \oplus \cdots \oplus Q_\theta(x)_L$$

This makes a comparison between the full state space transition probabilities and per-site factorized transition probabilities more convenient

$$P(\cdot \mid x, t) = \left( e^{t\mathbf{Q}} \right)_{x, \cdot} \tag{11}$$

$$p_\theta(\cdot \mid x, t) = \left( e^{t\mathbf{Q}_\theta(x)} \right)_{x, \cdot} \tag{12}$$

Now, we can uniformize both transition kernels by picking $\lambda = \max_z\{-\mathbf{Q}_{z,z}\}$ (which is also equal to $\max_z\{-\mathbf{Q}_\theta(x)_{z,z}\}$) and defining embedded DTMCs

$$R = I + \mathbf{Q}/\lambda \tag{13}$$

$$S = I + \mathbf{Q}_\theta(x)/\lambda \tag{14}$$

Uniformization tells us

$$e^{t\mathbf{Q}} = \sum_{n=0}^{\infty} e^{-\lambda t} \frac{(\lambda t)^n}{n!} R^n, \quad e^{t\mathbf{Q}_\theta(x)} = \sum_{n=0}^{\infty} e^{-\lambda t} \frac{(\lambda t)^n}{n!} S^n,$$

Using the matching assumption for single-site mutants stated in the theorem, we know that

$$\mathbf{Q}_{x, \cdot} = \mathbf{Q}_\theta(x)_{x, \cdot} \implies R_{x, \cdot} = S_{x, \cdot}.$$

Therefore, the $n = 0$ and $n = 1$ terms in the uniformization series match exactly for row $x$, meaning the differences start only at $n \geq 2$

$$\left( e^{t\mathbf{Q}} - e^{t\mathbf{Q}_\theta(x)} \right)_{x, \cdot} = \sum_{n=2}^{\infty} e^{-\lambda t} \frac{(\lambda t)^n}{n!} (R^n - S^n)_{x, \cdot}.$$

Because $(R^n)_{x, \cdot}$ and $(S^n)_{x, \cdot}$ are valid probability distributions, $\|(R^n - S^n)_{x, \cdot}\|_1 \leq 2$, allowing us to bound the $L_1$ error between the true transition probability vector and per-site factorized probability vector as

$$\|P(\cdot \mid x, t) - p_\theta(\cdot \mid x, t)\|_1 = \left\| \left( e^{t\mathbf{Q}} - e^{t\mathbf{Q}_\theta(x)} \right)_{x, \cdot} \right\|_1 \leq 2 \sum_{n=2}^{\infty} e^{-\lambda t} \frac{(\lambda t)^n}{n!} = 2 \left( 1 - e^{-\lambda t}(1 + \lambda t) \right)$$

Recognizing that the series $1 - e^{-u}(1 + u) = u^2/2 - u^3/6 + \cdots \leq u^2/2$ for $u \geq 0$ allows us to simplify the bound to[*]

$$\|P(\cdot \mid x, t) - p_\theta(\cdot \mid x, t)\|_1 \leq (\lambda t)^2 = O(t^2)$$

$\square$

### D.2. Exactness of Gillespie Sampling

**Lemma D.2.** *(Proof of Lemma 4.2)* *Let* $x_0, \ldots, x_{t_{N-1}}, x_{t_N}$ *be the trajectory of sequences sampled from the Gillespie procedure in Algorithm S1, using branch length $t$ and starting sequence $x_0$. For all $x \in \{x_0, \ldots, x_{N-1}\}$, assuming that*

$$(Q_\theta(x)_\ell)_{x_\ell, y_\ell} = \mathbf{Q}_{x,y}$$

*holds for all sequences $y$ with Hamming distance $d(x, y) = 1$, then $x_{t_N} \sim P(\cdot \mid x_0, t)$*

*Proof.* The proof follows from the fact that a continuous-time Markov chain is uniquely characterized by its holding time distributions and jump chain probabilities. By construction, the algorithm computes the total exit rate $\lambda_x \leftarrow -\sum_{\ell=1}^L Q_\theta(x)_{x_\ell, x_\ell}$ to sample the holding time. Under the lemma's condition that $(Q_\theta(x)_\ell)_{x_\ell, y_\ell} = \mathbf{Q}_{x,y}$ for all single residue mutants $y$, this sum is exactly equal to the target exit rate $-\mathbf{Q}_{x,x}$. Subsequently, the algorithm selects the next state $y$, corresponding to mutation $(\ell^*, a^*)$, with probability $P(\ell, a) = (Q_\theta(x)_\ell)_{x_\ell, a}/\lambda_x$. Due to the previous set of assumptions, this is identical to the transition probability $\mathbf{Q}_{x,y}/ - \mathbf{Q}_{x,x}$ of the target process. Since both the exponential holding times and discrete transition probabilities match those defined by the generator $\mathbf{Q}$ at every step, the simulated trajectory is a statistically exact realization of the true process, ensuring $x_{t_N}$ is distributed according to $P(\cdot \mid x_0, t)$. $\square$

### D.3. Relation Between Fixation Probability and Relative Fitness

Following Kimura (1962), we define $P_{\text{fix}}$ for a haploid population (which we use to describe affinity maturation due to asexual reproduction of B cells and allelic exclusion, which ensures that only one allele is expressed per B cell) as

$$P_{\text{fix}}(x \to y) = \frac{1 - e^{-2s_{xy}}}{1 - e^{-2N_e s_{xy}}}, \tag{15}$$

where $x$ is the wildtype allele, $y$ is a newly introduced allele, $s_{xy} = F_y - F_x$ is the selective advantage of allele $y$ over allele $x$, and $N_e$ is the effective population size. Crucially, the fixation probability expression is monotonic with respect to $s_{xy}$, which explains why the selection score calculated by Equation (5) shows strong Spearman correlation with empirical relative fitness measurements.

---

[*]The constant $\lambda$ corresponds to the maximum total substitution rate out of any sequence state. Under standard sequential point-mutation models, this rate scales at most linearly with sequence length and is bounded by the sum of per-site mutation rates.

