# OpenReview forum: "Conditionally Site-Independent Neural Evolution of Antibody Sequences"
_ICML.cc/2026/Conference — ICML 2026 regular_

### Official Review · Reviewer_5u4G · 2026-03-11

**Soundness:** 3
**Presentation:** 3
**Significance:** 2
**Originality:** 2
**Overall Recommendation:** 4
**Confidence:** 3

**Summary:**

This paper proposes COSiNE, a neural CTMC model for antibody affinity maturation. The model predicts site-specific rate matrices conditioned on the ful sequence and uses their matrix exponentials to define a factorized transition probability over evolved sequences. This construction is proved to give a first-order approximation to the sequential point-mutation process. COSiNE shows strong results on zero-shot variant effect prediction and guided antibody optimization via Guided Gillespie sampling.

**Compliance With Llm Reviewing Policy:**

Affirmed.

**Final Justification:**

The authors did a good work in the rebuttal and the paper has its value. I'm raising my score.

**Key Questions For Authors:**

1. Guided Gillespie sampling may be computationally expensive. Could the authors provide a computational cost analysis, such as how does the runtime scale with sequence length or sampling steps?
2. How sensitive are the zero-shot VEP results to the fixed choice of $t=0.2$? How robust is the performance under other branch-length settings?
3. Could the authors provide analysis on how does COSiNE perform when applied to longer branches or noisier clonal tree reconstructions?

**Limitations:**

yes

**Strengths And Weaknesses:**

# Strength
1. The paper tackles an important and well-motivated problem and reports promising empirical results
2. The paper does not merely present COSiNE as a purely heuristic neural model, but provide a first-order approximation proof with bounded error.

# Weakness
1. The approximation error is $O(t^2)$ in branch length so the theoretical guarantee is only compelling in sufficiently short-branch regimes and leaves a substantial gap between theory and realistic practice.
2. The method’s novelty appears somewhat limited, as the main design choice is to use a neural network to parameterize a context-dependent rate matrix. The paper does not sufficiently justify this design choice relative to a discrete flow matching (DFM) framework, which already provides an elegant way to model time-aware single-mutation dynamics, and no direct benchmark against DFM is provided.
3. The antibody optimization results are evaluated primarily through the model’s guided predictor and auxiliary plausibility measures such as humanness and pLDDT, but the paper does not validate improved affinity using stronger external biophysical evidence such as docking or energy-based metrics. This raises the concern that the method may be optimizing predictor score rather than true binding affinity.

---

> ### Author Rebuttal · Authors · 2026-03-31
>
> Thank you for taking the time to read our paper and for providing valuable feedback. We appreciate your questions and have provided detailed answers below.
>
> > Guided antibody optimization results are not validated using stronger external biophysical evidence like energy-based metrics.
>
> We directly address this concern in our response to Reviewer `2JJG`. In brief, we validate CoSiNE-optimized variants using DiffAffinity, a structure-based $\Delta\Delta G$ predictor trained on SKEMPI2 that outperforms established energy-based methods like FoldX and Rosetta — and is fully orthogonal to our sequence-based guidance predictor. We found that CoSiNE with guidance ($\gamma=1$) achieves the best binding affinity (lowest $\Delta\Delta G$) across all methods, while unguided CoSiNE is also competitive, suggesting our learned substitution rates capture meaningful binding constraints independently of the predictor.
>
> > How does the computational cost of *Guided Gillespie* scale by sequence length $L$ and by number of sampling steps $N$?
>
> As a brief reminder, *Guided Gillespie* utilizes our approximate TAG guidance (Equation 9), which leverages a Taylor-Series expansion to require only a single predictor call per sampling step regardless of sequence length. Empirically, it scales linearly in sequence length (due to the predictor forward pass) and linearly with the number of sampling steps.
>
> Below we fix the number of sampling steps at 50, noting that the guided runtime increases linearly with $L$ while unguided sampling remains constant because there are no calls to the predictor.
>
> | Sequence Length  | Guided Runtime (seconds) | Unguided Runtime (seconds) |
> | - | - | - |
> | $50$  | $5.16$ | $3.54$ |
> | $100$ | $5.94$ | $3.51$ |
> | $250$ | $8.53$ | $3.51$ |
> | $500$ | $12.2$ | $3.65$ |
>
> Below we fix the sequence length at 100, noting that both runtimes scale linearly with sampling steps.
>
> | Sampling Steps | Guided Runtime (seconds) | Unguided Runtime (seconds) |
> | - | - | - |
> | $5$   | $0.612$ | $0.356$ |
> | $10$  | $1.22$  | $0.734$ |
> | $25$  | $3.07$  | $1.76$  |
> | $50$  | $6.14$  | $3.50$  |
> | $100$ | $12.2$  | $6.96$  |
>
>
> > Sensitivity of VEP to the choice of branch length
>
> We show that CoSiNE's zero-shot VEP performance is robust to a range of $t$ values, and we've updated the manuscript to discuss this. Please see our response to `2JJG` for details.
>
> > The approximation error is in branch length so the theoretical guarantee is only compelling in sufficiently short-branch regimes and leaves a substantial gap between theory and realistic practice.
>
> We agree that our theoretical guarantees apply primarily to the short-branch regime, but this aligns with our focus on affinity maturation, which is a rapid process yielding short branches. While approximation errors exist over longer branches, our strong performance in zero-shot VEP and guided optimization demonstrate practical utility. Moreover, we note that classical models like WAG and LG remain widely successful in phylogenetic inference despite making stronger approximations (e.g. complete site independence).
>
> > Comparison of CoSiNE to discrete flow matching (DFM).
>
> While both CoSiNE and DFM use continuous-time Markov chains, they solve fundamentally different problems. DFM is a generative model mapping noise to an i.i.d. data distribution over an artificial algorithmic time ($\tau\in[0,1]$). Conversely, CoSiNE models actual phylogenetic evolution, learning a transition kernel between protein sequences over real biological time ($t>0$). Because DFM is not designed to model realistic state-to-state transitions over arbitrary time intervals, evaluating DFM on evolutionary tasks like affinity maturation would require substantial adaptation, which we consider beyond the scope of this paper but plan to discuss in the revised manuscript.
>
> > How does COSiNE perform on longer branches or noisier clonal tree reconstructions?
>
> To evaluate robustness to both longer branches and noisier tree reconstructions, we trained CoSiNE on the TrRosetta [1] dataset (~15k general protein families). These families span far greater evolutionary distances than antibody clonal trees, and their phylogenies are reconstructed from sequence homology rather than direct lineage tracing, making them inherently noisier. We compare against two classical substitution models, using the WAG and LG likelihood forms to fit new exchangeability matrices from scratch on the same TrRosetta training data. CoSiNE achieves lower test NLL than both baselines across all splits, indicating that context-dependent rate modeling reduces misspecification even under these more challenging conditions.
>
> | Model | Val NLL | Test NLL |
> |---|---|---|
> | WAG | 1.1702 | 1.2004 |
> | LG (4 rate categories) | 1.1390 | 1.1682 |
> | CoSiNE | **1.0051** | **1.0274** |
>
> [1] Improved protein structure prediction using predicted interresidue orientations. Yang, J. et al. PNAS 2020

---

> > ### Author Rebuttal · Reviewer_5u4G · 2026-04-02
> >
> > Thank you for the detailed rebuttal. The additional analyses improve the empirical support for the paper. However, the paper’s biological significance still depends on stronger experimental validation, and the method’s novelty remains a key limitation.

---

> > > ### Author Response · Authors · 2026-04-05
> > >
> > > Thank you for your continued feedback. We agree that wet-lab validation remains necessary for guided generation, and we recognize this as an important direction for future work.
> > >
> > > That said, we believe the current empirical evidence is strong. CoSiNE achieves state-of-the-art zero-shot VEP across 9 assays spanning diverse antibodies, antigens, and experimental platforms. For guided generation, under matched oracle access, Guided Gillespie outperforms all alternative guidance schemes, and the DiffAffinity validation confirms that improvements transfer to an orthogonal structure-based metric.
> > >
> > > Regarding novelty, the neural CTMC architecture, selection score decomposition, and Guided Gillespie sampling each represent methodological contributions. In particular, CoSiNE is, to our knowledge, the first model to combine the expressivity of deep neural networks with the phylogenetic structure of affinity maturation, enabling both context-dependent rate estimation and principled sampling via Gillespie dynamics. We acknowledge the connection to DFM but maintain that the two frameworks solve fundamentally different problems, as discussed in our rebuttal.
> > >
> > > We hope the additional experiments address your concerns and that you might reconsider your assessment. We appreciate your engagement with our work.

---

### Official Review · Reviewer_EU4y · 2026-03-12

**Soundness:** 3
**Presentation:** 3
**Significance:** 3
**Originality:** 3
**Overall Recommendation:** 4
**Confidence:** 3

**Summary:**

CoSiNE is a conditionally site-independent neural CTMC model for antibody affinity maturation. The idea is to predict site-specific rate matrices conditioned on the full parent sequence, giving you a factorized transition model that is tractable but still captures context dependence and epistasis through the neural parameterization. The paper shows a first-order approximation linking this to sequential single-point mutation processes, proposes Gillespie sampling to better respect single-mutation dynamics, and adds a guided Gillespie scheme for target-conditioned optimization. Experiments cover clonal-tree transition modeling, zero-shot variant effect prediction, and in silico guided antibody optimization.

**Compliance With Llm Reviewing Policy:**

Affirmed.

**Final Justification:**

I thank the authors for the constructive rebuttal. The additional analyses were helpful: the new VEP results, the ESM2 ablation, the clarification of paired vs. single-chain conditioning, and the sharper framing of the optimization setting all strengthen the paper and make its scope clearer.

My main reservation remains the same as in the original review: the theoretical justification relies on learned rates approximating true instantaneous rates, but the current training procedure does not directly establish that. I therefore still view the theoretical connection as a useful motivation rather than a full guarantee for the learned model. I also continue to see the optimization results as proof-of-concept and the empirical validation as somewhat limited in breadth.

Overall, the rebuttal partially addressed my concerns and reinforced my view that this is a worthwhile paper with clear limitations. My final recommendation therefore remains weak accept.

**Key Questions For Authors:**

1. How sensitive are the VEP results to the chosen fixed branch length $t=0.2$ for all assays?
2. Can the authors provide a more direct evaluation of whether learned rates approximate instantaneous rates, beyond synthetic codon experiments?
3. How much of the performance gain comes from the pretrained ESM2 backbone versus the CoSiNE transition parameterization itself?
4. Why does paired conditioning underperform single-chain conditioning on more than half of the Table S2 datasets?
5. Do the authors intend CoSiNE mainly as a model of affinity maturation, or as a general model of protein evolution? The current framing mixes both.

**Limitations:**

yes

**Strengths And Weaknesses:**

Strengths
1. The problem formulation is well-grounded. Current antibody language models treat observed sequences as i.i.d., but affinity maturation is a lineage-structured evolutionary process. Pointing out this mismatch and building a model around it is one of the paper's clearest contributions.
2. The modeling idea — site-wise matrix exponentials with context-conditioned rate matrices (Eq. 2) — is a nice bridge between phylogenetics and deep sequence models. It keeps likelihood computation tractable while relaxing classical site-independence through whole-sequence conditioning.
3. The first-order approximation result is useful: if single-step rates match the sequential point mutation generator, transition probability error is quadratic in branch length. This gives a principled justification for why the approximation should hold on the short branches typical of affinity maturation.
4. Gillespie sampling is a natural fit here and the paper backs it up empirically. Both synthetic experiments and real-antibody proxy analyses show it outperforms naive factorized sampling from matrix exponentiation.
5. VEP results are strong. On FLAb2-derived DMS assays, CoSiNE is competitive or best across nearly all datasets, and the SHM-corrected selection score adds measurable value over raw transition likelihood.

Weaknesses

1. The theory assumes learned rates match true instantaneous rates, but training on the approximate factorized likelihood doesn't guarantee this. The authors acknowledge that long branches introduce bias from marginalizing out intermediates, and their proposed SNR-weighting doesn't help on real data. So the theoretical guarantees apply to an idealized setting, not the model actually used in experiments.
2. VEP results are encouraging but narrow. The benchmark covers only a few DMS assays, some quite small. Not enough to conclusively establish broad superiority.
3. Guided optimization uses weak external oracles and is evaluated purely in silico. The paper notes that stronger guidance can exceed known biological binders, likely from oracle exploitation. This is better read as proof-of-concept than a validated design pipeline.
4. Evidence for paired heavy/light conditioning is mixed. Table S2 shows single-chain conditioning outperforms paired on four of seven datasets. The margins are small, but this complicates the claimed importance of cross-chain context.

---

> ### Author Rebuttal · Authors · 2026-03-31
>
> Thank you for your detailed review of our paper. We appreciate your thoughtful questions and have addressed them thoroughly below. We hope this response clarifies our approach and that the reviewer might consider further strengthening their support for our work.
>
> > Sensitivity of VEP to the choice of branch length
>
> We show that CoSiNE's VEP performance is robust to a range of $t$ values, and we've updated the manuscript to discuss this. Please see our response to `2JJG` for details.
>
> > Additional VEP results
>
> We have evaluated CoSiNE on 2 new assays, where CoSiNE outperforms all other methods. Please see our response to `8UQx` for details.
>
> > Validation of guided antibody optimization with external oracle.
>
> To address the concern about weak oracles, we validate our optimized variants using DiffAffinity, a structure-based $\Delta\Delta G$ predictor trained on general PDB complexes (details in our response to Reviewer `2JJG`). In brief, CoSiNE ($\gamma=1$) achieves the best binding affinity (lowest $\Delta\Delta G$) across all methods, while methods that overly exploit the guidance predictor do poorly. All in all, we agree that wet-lab validation remains the gold standard and have modified our claims accordingly.
>
> > Source of CoSiNE Performance Gain
>
> To address this question, we ablated the ESM2 backbone by training a version of CoSiNE from scratch (8 million parameters, random weight initialization) using the same training dataset to understand how important the pretrained ESM2-150 component is for VEP.
>
> #### Expression Datasets
>
> | Model | Koenig (H) | Koenig (L) | Adams |
> | :--- | :--- | :--- | :--- |
> | **DASM** | *0.596* | 0.474 | 0.270 |
> | **ESM2-650M** | 0.326 | 0.429 | 0.124 |
> | **ProGen2-Small** | 0.407 | **0.513** | -0.024 |
> | **CoSiNE-ESM2** | **0.613** | *0.508* | **0.464** |
> | **CoSiNE-8M** | 0.503 | 0.474 | *0.447* |
>
> #### Binding Datasets
>
> | Model | Koenig (H) | Koenig (L) | Shaneh. (119) | Shaneh. (120) |
> | :--- | :--- | :--- | :--- | :--- |
> | **DASM** | *0.415* | 0.327 | 0.450 | **0.536** |
> | **ESM2-650M** | 0.063 | 0.265 | 0.227 | 0.360 |
> | **ProGen2-Small** | 0.098 | *0.332* | 0.119 | 0.070 |
> | **CoSiNE-ESM2** | **0.456** | **0.371** | *0.498* | **0.536** |
> | **CoSiNE-8M** | 0.363 | 0.330 | **0.508** | *0.531* |
>
> **Bold** for best, *Italics* for second best
>
> The ESM2 backbone improves the CoSiNE selection score's correlation with fitness on all but 1 dataset. But, the 8M version still performs exceptionally well — the average $\Delta\rho$ between the ESM2 version and 8M version is only 0.041. Therefore, the ablation suggests that the pretrained ESM2 backbone provides a useful initialization, but is not the primary source of predictive power.
>
> Together with the WAG comparison (response to 8UQx), which shows context-dependent rate estimation matters, and the ESM2-150M baseline in Table 1, which shares CoSiNE's architecture but was trained as a standard language model, these results suggest that the context-aware evolutionary training objective is the primary driver of CoSiNE's VEP performance.
>
> > Paired vs Single Chain Conditioning
>
> We agree that the paired versus single conditioning results are mixed on a per-dataset basis. However, as shown in Table S2, the gains from paired conditioning when it helps are substantial (average +0.121 in Spearman correlation), while the losses when single conditioning is better are negligible (average +0.006). This asymmetry suggests that inter-chain interactions are genuinely informative for some fitness landscapes but not others, rather than paired conditioning being uniformly helpful or harmful.
>
> We note that the majority of our training data consists of unpaired single-chain sequences, which likely limits the model's ability to fully leverage paired context. We expect that as more paired heavy-light sequencing data becomes available, the benefits of cross-chain conditioning will become more consistent. We have updated the manuscript to discuss this nuance more carefully.
>
> > Approximation of Instantaneous Rates and Framing of Contribution
>
> Regarding the evaluation of learned rates: directly validating instantaneous rates requires ground-truth knowledge of the generative process, which is only available in simulation — hence our synthetic codon experiment. We refer the reviewer to Appendix C.3, which analyzes test-set sampling error in the real antibody setting, providing additional evidence that our learned rates are well-calibrated. Regarding scope: CoSiNE is intended primarily as a model of antibody affinity maturation; extending to indels could yield a general protein evolution simulator, but this is left as future work. We agree the current framing conflates these goals and have clarified this in the revised manuscript.

---

> > ### Author Rebuttal · Reviewer_EU4y · 2026-04-02
> >
> > The authors provided a constructive and useful rebuttal, and several of my questions were partially addressed. In particular, the added ablations and additional VEP results are helpful, and I appreciate the clarification of claims around the optimization setting. That said, my primary concern remains: the paper’s theoretical justification depends on learned rates approximating the true instantaneous rates, but the current training procedure does not directly establish this. Since this issue goes to the core modeling justification, I do not think it can be fully resolved within the scope of a short rebuttal. I also still find the empirical validation somewhat limited in breadth, especially for broader VEP/generalization claims, and I continue to view the optimization results as proof-of-concept rather than a fully validated design pipeline. For these reasons, I view the concerns as partially resolved.

---

> > > ### Author Response · Authors · 2026-04-05
> > >
> > > Thank you for your continued feedback. We're glad we were able to address some of your concerns with our rebuttal, but respect your remaining concerns.
> > >
> > > While it's true that training on the factorized likelihood (Equation 2) does not guarantee recovery of true instantaneous rates, we note that this is an inherent tradeoff of making inference tractable over the full sequence state space. Our synthetic codon experiments (Section C.1) demonstrate that the estimated rates closely approximate the ground truth in controlled settings, and the state-of-the-art empirical performance across 9 VEP assays and the DiffAffinity validation suggest that the learned rates are sufficiently well-calibrated for practical downstream tasks. We view closing this theory-practice gap as an important direction for future work.
> > >
> > > We agree that experimental measurements of design sequences would be necessary to fully validate our design pipeline. However, within computational evaluation, our DiffAffinity results show that CoSiNE's guided designs are structurally viable and outperform alternative schemes under matched oracle access, which presents a significant contribution to the field.
> > >
> > > Nonetheless, we really appreciate that you've clearly engaged deeply with the technical details of our work, and we think your assessment reflects genuine expertise on these questions. Given the depth of your engagement, we would appreciate an update to your confidence score that reflects your familiarity with the work.

---

### Official Review · Reviewer_8UQx · 2026-03-13

**Soundness:** 3
**Presentation:** 3
**Significance:** 3
**Originality:** 3
**Overall Recommendation:** 3
**Confidence:** 3

**Summary:**

This paper presents COSINE, a neural continuous-time Markov chain for modeling antibody sequence evolution during affinity maturation. The method combines the temporal structure of phylogenetic substitution models with neural, context-dependent rate parameterization, and derives a first-order approximation to sequential point-mutation dynamics. The model is trained by likelihood maximization on clonal-tree transitions and extended with classifier-guided Gillespie sampling for inference-time steering. Empirically, the paper reports improved zero-shot antibody variant-effect prediction and promising in silico affinity-optimization results.

**Compliance With Llm Reviewing Policy:**

Affirmed.

**Key Questions For Authors:**

1. How sensitive are the VEP results to branch-length calibration, both globally and per assay? Stable gains under reasonable alternatives would increase confidence in robustness.
2. Can the authors report per-assay results or paired tests against the strongest baselines, rather than only aggregate correlations? This would show whether the gains are consistent or dataset-specific.
3. How much of the gain comes from context-dependent epistasis modeling, versus evolutionary supervision or model capacity more broadly? Clearer attribution would strengthen the central claim.
4. For Guided Gillespie, are compute budgets and oracle-access patterns matched to the optimization baselines? A tightly controlled comparison is needed to assess this contribution.

**Limitations:**

1. The empirical evidence is limited to antibody substitution settings.
2. Some conclusions depend on branch-length calibration choices.
3. The optimization results are shown only in oracle-guided in silico settings, so claims about practical design utility should be scoped accordingly.

**Strengths And Weaknesses:**

Strengths:
1. The CTMC formulation is well motivated, and the first-order approximation gives the method a clear technical foundation.
2. The VEP evaluation is reasonable, with useful supporting analyses on estimation error, runtime, and branch-length effects in the appendix.
3. Modeling evolutionary transitions rather than only sequence marginals is a meaningful and original contribution for antibody modeling.

Weaknesses:
1. The empirical evidence is still narrow, centered on a small number of antibody DMS datasets.
2. Key choices such as branch-length calibration, score construction, and the compute-quality trade-off of Guided Gillespie are not fully clarified in the main text.
3. The optimization results are entirely oracle-guided and in silico, so broader design claims should be stated more cautiously.

---

> ### Author Rebuttal · Authors · 2026-03-31
>
> Thank you for your feedback. We appreciate the time and effort that you put into reviewing our work. Below, we address your questions and comments in detail.
>
> > Clarification of VEP correlation calculation
>
> Regarding the question on providing per-assay results rather than aggregate correlations, our current VEP results (Table 1) already show per-assay results, making CoSiNE directly comparable to the existing baselines. We also provide a paired evaluation for modeling held-out clonal trees against the most directly comparable baseline (DASM) in Figure 2. Please feel free to clarify if we are misinterpreting your question.
>
> > Additional VEP results
>
> The table shows Spearman correlations for **2 new VEP benchmarks** measuring binding affinity ([1]). We chose these assays because they use distinct antibodies, antigens, and experimental technologies (MAGMA-seq) from those in Table 1.
>
> | Model | Petersen (319-345) | Petersen (222-1C06) |
> | :--- | :--- | :--- |
> | **AbLang-2** | 0.199 | 0.060 |
> | **DASM** | *0.395* | *0.286* |
> | **ESM2-150M** | 0.250 | -0.139 |
> | **ESM2-650M** | 0.278 | 0.013 |
> | **ProGen2-Small** | 0.329 | 0.024 |
> | **ProGen2-Medium** |0.294  | 0.036 |
> | **CoSiNE ($t=0.2$)** | **0.504** | **0.328** |
>
> **Bold** for best, *Italics* for second best
>
> CoSiNE outperforms all baselines, consistent with previous results.
>
> [1] Petersen et al. (2024) An integrated technology for quantitative wide mutational scanning of human antibody fab libraries Nat. Commun 15:3974
>
>
> > Sensitivity of VEP to the choice of branch length
>
> Please see our response to reviewer `2JJG`, where we find that CoSiNE indeed makes stable gains over baseline methods for a wide range of reasonable $t$ values.
>
> > Validation of guided antibody optimization with external oracle.
>
> We have addressed this concern in our response to Reviewer `2JJG`, where we validate CoSiNE-optimized variants using DiffAffinity, a structure-based $\Delta\Delta G$ predictor trained on PPI complexes in SKEMPI2. We found that CoSiNE with guidance ($\gamma=1$) achieves the best binding affinity (lowest $\Delta\Delta G$) across all methods, while unguided CoSiNE is also competitive, suggesting our learned substitution rates capture meaningful selection constraints independently of the predictor. We agree that wet-lab validation remains the gold standard and have moderated our claims accordingly.
>
> > How much of the gain comes from context-dependent epistasis modeling, versus evolutionary supervision or model capacity more broadly?
>
> We investigate the importance of context-dependence by comparing CoSiNE against a classical context-independent substitution model: a single 20×20 exchangeability matrix fit using the WAG likelihood form. We estimate these parameters via MLE on the same training transitions used for CoSiNE, ensuring a fair comparison. We omit further implementation details for brevity, but are happy to elaborate upon request.
>
> On our test set, CoSiNE achieves significantly lower NLL per token than WAG across all branch lengths, with the gap widening for longer branches. This suggests that sequence-context conditioning allows CoSiNE to capture epistatic interactions that accrue over long evolutionary distances.
>
> | Branch Length    | Num. Mutations | WAG NLL | CoSiNE NLL |
> | -- | -- | -- | -- |
> | $[0, 0.01)$      | $1.10\pm0.307$ | $0.0717$ | $0.0663$ |
> | $[0.01, 0.05)$   | $3.20\pm1.81$ | $0.171$   | $0.147$  |
> | $[0.05, 0.15)$   | $10.1\pm4.22$ | $0.454$   | $0.378$  |
> | $[0.15, \infty)$ | $20.6\pm5.45$ | $0.836$   | $0.676$  |
>
> Please also refer to our response to reviewer `EU4y` where we ablate the ESM2 backbone by training a smaller CoSiNE model from scratch. We will provide a discussion of all these experiments in the main text.
>
> > Clarification of key choices in the main text (branch length calibration, score construction, compute-quality tradeoff of Guided Gillespie)
>
> We already discuss selection score construction in Section 4.1, but will discuss VEP branch length calibration (see response to `2JJG`) and compute-quality tradeoff (see Appendix C.7) in the main text.
>
> > Clarification of controlled oracle budget in CDR optimization experiment
>
> In the local CDR optimization experiment (Section 5.5), we restricted all methods to a maximum of 5 oracle calls per optimized sequence and report this metric in the $N_{\text{oracle}}$ column of Table S4. We will state this explicitly in the main text.

---

> > ### Author Rebuttal · Reviewer_8UQx · 2026-04-03
> >
> > Thank you for your replies.

---

> > > ### Author Response · Authors · 2026-04-03
> > >
> > > Thank you so much for your time and for confirming that your concerns have been fully resolved! We are very glad that the additional experiments and clarifications addressed the points you raised. **We kindly ask if you might consider updating your initial score to reflect this.** We deeply appreciate your constructive feedback, which has genuinely strengthened our paper.

---

### Official Review · Reviewer_2JJG · 2026-03-16

**Soundness:** 3
**Presentation:** 3
**Significance:** 3
**Originality:** 3
**Overall Recommendation:** 4
**Confidence:** 5

**Summary:**

This paper introduces COSINE, a framework that reconciles deep learning expressivity with classical phylogenetic dynamics by parameterizing a continuous-time Markov chain with a neural backbone to model antibody affinity maturation. By predicting site-specific rate matrices conditioned on the full sequence, the model captures complex epistatic interactions that independent-site models miss. The authors provide a solid theoretical grounding by proving COSINE is a first-order approximation of sequential point mutation processes with a quadratic error bound relative to branch length. For functional design, the study introduces Guided Gillespie sampling accelerated by Taylor-approximated guidance (TAG), achieving up to a 928x speedup over exact methods without sacrificing performance. Empirically, COSINE delivers state-of-the-art zero-shot variant effect prediction by disentangling selection from mutational bias and successfully optimizes antibody binding while maintaining structural plausibility and humanness.

**Compliance With Llm Reviewing Policy:**

Affirmed.

**Final Justification:**

my concerns are Partially resolved. I maintain my score.

**Key Questions For Authors:**

None

**Strengths And Weaknesses:**

Strengths

The paper is technically sound, providing rigorous proofs and solid results against strong baselines. The presentation is clear, effectively bridging the gap between evolutionary biology and deep learning. Its primary significance lies in the massive speedup offered by the guidance approach, making it highly practical for real-world lab applications. The originality is reflected in the creative use of neural rate matrices to capture complex site dependencies within a global context.

Weaknesses

The method’s soundness is limited by an approximation error that naturally increases over longer evolutionary distances. In terms of significance, the current framework cannot handle insertions or deletions, restricting it to fixed-length sequences. Additionally, the reliance on predominantly single-chain training data may hinder the model's ability to fully capture paired-chain interactions.

---

> ### Author Rebuttal · Authors · 2026-03-31
>
> Thank you for your encouraging comments about our work. We are pleased that the reviewer recognizes the practical significance of our guided sampling approach and the originality of our neural CTMC parameterization. To strengthen our contribution, we provide some additional results of interest.
>
> > Validation of guided antibody optimization with external oracle.
>
> To address concerns about external validation, we evaluate CoSiNE-optimized antibody variants using DiffAffinity [1], a structure-based $\Delta\Delta G$ predictor. We follow a slightly modified protocol from Section 5.5: we allow up to 3 mutations in the heavy chain variable region and sample 100 variants per method.
>
> **Why DiffAffinity is an appropriate oracle:** DiffAffinity is trained on SKEMPI2 binding affinity data using structural side-chain representations, making it fully orthogonal to our sequence-based guidance predictor trained on neutralizing antibodies in CoVAbDab. The authors validated DiffAffinity specifically on the same SARS-CoV-2 complex (PDB: 7FAE) used in this experiment, where it outperforms FoldX and Rosetta.
>
> **Results:** CoSiNE with guidance ($\gamma=1$) achieves the lowest $\Delta\Delta G$ (strongest improvement in binding affinity) and the most variants with $\Delta\Delta G<0$ (i.e. binding more tightly than wild-type) across all methods. Methods that over-optimize the neutralization predictor (genetic algorithm, stronger guidance) score poorly due to epistemic uncertainty in the guidance model. Notably, unguided CoSiNE is also competitive (34 variants with $\Delta\Delta G<0$, second highest), suggesting that our learned substitution rates capture meaningful selection constraints independently of the predictor.
>
> | Method | Median $\Delta F$ | Min $\Delta\Delta G$ | $\Delta\Delta G<0$ |
> | -- | -- | -- | -- |
> | Random                | $-0.0453$ | $-1.09$  | $31$ |
> | CoSiNE (unguided)     | $0.0185$  | $-1.53$  | $34$ |
> | Ablang (unguided)     | $0.0917$  | $-1.03$  | $22$ |
> | CoSiNE ($\gamma=1$)   | $0.112$   | $\textbf{-1.90}$  | $\textbf{38}$ |
> | Ablang ($\gamma=1$)   | $0.218$   | $-1.15$  | $23$ |
> | CoSiNE ($\gamma=5$)   | $0.594$   | $-1.65$  | $18$ |
> | Ablang ($\gamma=5$)   | $0.787$   | $-0.699$ | $13$ |
> | Genetic Algorithm     | $0.783$   | $-0.366$ | $23$ |
>
> [1] Predicting mutational effects on protein-protein binding via a side-chain diffusion probabilistic model. Liu et al. NeurIPS 2023.
>
> > Sensitivity of VEP results to the choice of branch length
>
> In the following tables we investigate the robustness of CoSiNE to the choice of $t$ for VEP tasks. We evaluate CoSiNE on all the assays from the paper using 6 branch length values ranging from $t=0.01$ to $t=10$. We report the Spearman correlation ($\rho$) for each CoSiNE model, **bolded** if it outperforms all non-CoSiNE baselines (this includes methods not shown below for space constraints, but whose results can be found in Table 1 of manuscript).
>
> #### Expression Datasets
>
> | Model | Koenig (H) | Koenig (L) | Adams |
> | :--- | :--- | :--- | :--- |
> | **DASM** | 0.596 | 0.474 | 0.270 |
> | **ESM2-650M** | 0.326 | 0.429 | 0.124 |
> | **ProGen2-Small** | 0.407 | 0.513 | 0.231 |
> | **CoSiNE t=0.01** | 0.370 | 0.349 | **0.284** |
> | **CoSiNE t=0.1** | 0.584 | 0.487 | **0.441** |
> | **CoSiNE t=0.2** | **0.613** | 0.508 | **0.464** |
> | **CoSiNE t=0.3** | **0.624** | 0.500 | **0.463** |
> | **CoSiNE t=0.4** | **0.605** | 0.509 | **0.425** |
> | **CoSiNE t=10** | 0.407 | 0.325 | **0.288** |
>
> #### Binding Datasets
>
> | Model | Koenig (H) | Koenig (L) | Shaneh. (119) | Shaneh. (120) |
> | :--- | :--- | :--- | :--- | :--- |
> | **DASM** | 0.415 | 0.327 | 0.450 | 0.536 |
> | **ESM2-650M** | 0.063 | 0.265 | 0.227 | 0.360 |
> | **ProGen2-Small** | 0.098 | 0.332 | 0.119 | 0.070 |
> | **CoSiNE t=0.01** | 0.240 | 0.217 | **0.486** | **0.537** |
> | **CoSiNE t=0.1** | **0.427** | **0.352** | **0.505** | **0.549** |
> | **CoSiNE t=0.2** | **0.456** | **0.371** | **0.498** | **0.536** |
> | **CoSiNE t=0.3** | **0.471** | **0.375** | **0.503** | 0.533 |
> | **CoSiNE t=0.4** | **0.472** | **0.378** | **0.505** | 0.530 |
> | **CoSiNE t=10** | 0.217 | 0.209 | **0.477** | 0.480 |
>
> The CoSiNE selection score is most reliable in a relatively large range from $t=0.1$ to $t=0.4$, where the maximum $\Delta\rho$ across settings is only $0.045$. Furthermore, each model outperforms all other baselines on at least 5/7 assays and would perform second best on the remaining 2. Performance unsurprisingly degrades at the extremes ($t=0.01$ and $t=10$), though correlations remain significantly positive and often competitive with baselines.
>
> One could certainly achieve better performance by calibrating choice of $t$ per assay, but we chose a fixed value of $t=0.2$ across all assays to avoid unfairly fitting to the VEP benchmarks, given that real application settings would be zero-shot and we wouldn't have access to true fitness values.

---

> > ### Author Rebuttal · Reviewer_2JJG · 2026-04-03
> >
> > the authors proactively added an orthogonal external oracle (DiffAffinity) to validate binding optimization, and provided a thorough sensitivity analysis on branch length showing robust performance across a wide range. so I hold my score.

---

> > > ### Author Response · Authors · 2026-04-05
> > >
> > > Thank you for your continued support and precise assessment of our work. We wanted to briefly summarize the new results from our responses to other reviewers that we believe strengthen the paper overall. We also noticed you selected the partially resolved status with potential follow-up questions; please don't hesitate to send them our way, as we would be more than happy to answer them.
> > >
> > > - We showed the robustness of CoSiNE's VEP performance to the choice of $t$ by sweeping over a wide range of possible values. Within a large range of $t=0.1$ to $t=0.4$, CoSiNE maintains its strong state-of-the-art performance, with a maximum delta of 0.045 in Spearman correlation between $t$ settings.
> > > - We ablated the ESM2 backbone by training an 8M-parameter CoSiNE from scratch. VEP performance drops only slightly (average Spearman gap of 0.041), confirming that the context-aware evolutionary training objective, not the pretrained backbone, is the primary source of predictive power.
> > > - We fit a WAG-style exchangeability matrix via MLE on our training data and show that CoSiNE achieves significantly lower test NLL across all branch lengths, with the gap widening at longer branches. This isolates the contribution of context-dependent rate modeling.
> > > - We trained CoSiNE on TrRosetta protein families, which span much greater evolutionary distances and use noisier reconstructed phylogenies. CoSiNE outperforms both WAG and LG baselines on held-out test NLL, directly addressing concerns about robustness to longer branches and noisy trees.
> > > - We added 2 new VEP assays (Petersen et al., 2024), bringing the total to 9 datasets across diverse antibodies, antigens, and experimental platforms. CoSiNE outperforms all baselines on both.
> > > - We provided computational scaling results for Guided Gillespie, showing linear scaling in both sequence length and sampling steps.
> > >
> > > We hope these additions help address the evaluation breadth concerns raised elsewhere in the review process.

---

### Decision · Program_Chairs · 2026-04-30

**Decision:**

Accept (regular)

**Comment:**

This paper proposes a novel bridge between phylogenetic modeling and deep learning via context-dependent CTMCs, with solid theory and strong VEP performance. The reviewers raise several concerns about the experimental settings, and the theory-practice gap. The authors largely address these in the rebuttal, and they should make sure to add the new results and arguments to the final version.